# Limited surface impacts of the January 2021 sudden stratospheric warming

N. A. Davis [1✉], J. H. Richter [2], A. A. Glanville[2], J. Edwards[2] & E. LaJoie[3]

Subseasonal weather prediction can reduce economic disruption and loss of life, especially during "windows of opportunity" when noteworthy events in the Earth system are followed by characteristic weather patterns. Sudden stratospheric warmings (SSWs), breakdowns of the winter stratospheric polar vortex, are one such event. They often precede warm temperatures in Northern Canada and cold, stormy weather throughout Europe and the United States - including the most recent SSW on January 5th, 2021. Here we assess the drivers of surface weather in the weeks following the SSW through initial condition "scrambling" experiments using the real-time CESM2(WACCM6) Earth system prediction framework. We find that the SSW itself had a limited impact, and that stratospheric polar vortex stretching and wave reflection had no discernible contribution to the record cold in North America in February. Instead, the tropospheric circulation and bidirectional coupling between the troposphere and stratosphere were dominant contributors to variability.

[1] Atmospheric Chemistry Observations and Modeling Laboratory, National Center for Atmospheric Research, Boulder, CO, USA. [2] Climate and Global Dynamics Laboratory, National Center for Atmospheric Research, Boulder, CO, USA. [3] NOAA/NCEP/Climate Prediction Center, College Park, MD, USA. ✉email: nadavis@ucar.edu

Subseasonal forecasting is an emerging frontier of Earth system prediction. Government and industry stakeholders have a substantial interest in both the average weather conditions and the potential for extreme weather at 10–30 day lead times[1,2]. The Subseasonal Experiment (SubX)[3] and the Subseasonal to Seasonal (S2S) Prediction Project[4] both aim to improve operational subseasonal forecasting by supporting research into its performance, especially during "windows of opportunity"[2] when lead times may be significantly enhanced.

SubX and S2S are collaborative efforts that maintain databases of subseasonal forecasts from different modeling agencies. Attributing sources of predictability using these databases can be challenging, as model configurations vary and causality is difficult to discern in any numerical simulation. Atmospheric initial conditions have the greatest influence on a forecast in the first 2 weeks after initialization, while ocean and land initial conditions become more important at longer lead times[5,6]. However, the contributions of the initial states of the atmosphere, ocean, sea ice, and land to the predictability of particular events are inadequately quantified[7].

In the troposphere—the lowest layer of the atmosphere - the timescale of variability is short. Locally, the sun heats Earth's surface which invigorates atmospheric convection and produces a diurnal cycle of temperature. On a global scale, differential solar heating maintains the equator-to-pole temperature gradient that powers waves governing weekly variations in weather. Unlike in the troposphere, variability in the stratosphere—far above convection - is driven by slower processes like seasonal heating and planetary-scale waves. As a result, the stratosphere can act as a source of extended predictability for the troposphere[8].

Approximately, once every two years in the Northern Hemisphere during winter, breaking planetary-scale waves rapidly warm the polar stratosphere over several days and split or displace the stratospheric polar vortex—a sudden stratospheric warming (SSW)[9,10]. SSWs can be driven by enhanced wave energy propagating upward from the troposphere[11–13], sometimes produced by atmospheric blocking[14]; by waves generated at the boundary between the troposphere and the stratosphere[15,16]; and by variations in the stratosphere that focus and enhance the effect of otherwise normal wave energy[17–21]. While they are dynamically forced events that occur over a matter of days, they can be skillfully predicted even at the seasonal timescale[22].

In the month following an SSW there is generally anomalous warmth over Northern Canada and Alaska, the Middle East, and Central Asia, anomalous cold over Siberia and Northern Europe[23], and cold air outbreaks in Europe and the United States[23–26]. These patterns reflect the surface manifestation of the negative phase of the Northern Annular Mode (NAM)[27], which characterizes the latitudinal position of the midlatitude jet stream and appears to descend from the stratosphere to the troposphere during and after an SSW[28]. Surface predictability is enhanced in Asia, the Central United States, and the Middle East in the "window of opportunity" following an SSW, but it is degraded in Europe[23,29,30].

In spite of their broad impacts and scientific novelty, the mechanisms by which SSWs couple to surface weather and impact predictability are poorly understood[10]. The potential surface impacts of the recent SSW that occurred on January 5th, 2021 generated media interest[31,32] and an informative editorial by atmospheric scientists which discussed the difficulty in attributing individual weather events to SSWs[33].

Here, we provide an unambiguous evaluation of the surface impacts of the January 5th, 2021 SSW and the stratospheric variability in its aftermath using a series of experimental ensemble forecasts. In these experimental forecasts, we selectively leave different aspects of the atmosphere and Earth system uninitialized during the forecast spin-up, creating a "scrambled" initial state that directly tests the role of each aspect of the Earth system in driving the observed event. We show that forecasts initialized with scrambled stratospheric initial conditions explain most of the observed surface temperature variability in the month after the SSW and during the cold air outbreak. Differences in the circulation dynamics between these forecasts reveal that disturbed stratospheric states may be an important feedback on persistent tropospheric weather, rather than its proximate cause.

## Results

**Subseasonal prediction framework.** The CESM2(WACCM6) subseasonal Earth system prediction framework at the National Center for Atmospheric Research uses the Community Earth System Model version 2 in the Whole Atmosphere Community Climate Model version 6 configuration (CESM2(WACCM6))[34,35], which simulates atmospheric processes from the surface to approximately 140 km altitude. A 21-member ensemble forecast is generated every Monday and contributes to the National Oceanic and Atmospheric Administration's (NOAA) experimental week 3–4 outlooks.

Operational forecast models generally use data assimilation to create initial conditions from observations, while models intended for research often initialize directly from a meteorological reanalysis. Subseasonal forecast systems also typically initialize their ocean, sea ice, and land conditions as close to observations as possible, as the initial state of all of these components can be important for subseasonal prediction[36]. CESM2(WACCM6) is unique because it generates initial conditions for its atmosphere, ocean, sea ice, and land components by relaxing the atmosphere to a near-real-time reanalysis in the week before each initialization. This ensures atmospheric chemistry and other Earth system components are spun up to the atmospheric state at the time of initialization.

The technique can be modified to only relax either the stratosphere or the troposphere to the reanalysis, so that either the troposphere or the stratosphere, respectively, diverge from their observed state during spin-up and become "scrambled" at the initialization time. A simulation can also be initialized with an atmospheric state from a different year, for example, one in which the polar vortex is strong and an SSW did not occur, to assess the role of ocean, sea ice, and land forcing on the forecast. These experimental forecasts can be used to directly attribute forecasted surface temperatures to the SSW itself, to the prevailing tropospheric circulation concurrent with the SSW, or to the influence of surface boundary conditions.

Four ensemble forecasts with 21 members were initialized on Monday, January 4, 2021, 1 day prior to the SSW: a forecast with accurate initial conditions ("standard forecast"), a forecast with scrambled tropospheric initial conditions (and accurate stratospheric initial conditions), a forecast with scrambled stratospheric initial conditions (and accurate tropospheric initial conditions) in which no SSW occurs (see Supplementary Fig. 1), and a forecast with scrambled atmospheric initial conditions (with initial conditions set to January 2, 2017, a year with no major SSW). Most of the forecast quantities we examine are ensemble means over all 21 members.

We emphasize that the scrambled initial conditions are not random. During spin-up, the scrambling procedure simulates one-way coupling from the correctly-initialized portion of the atmosphere to the scrambled portion of the atmosphere. Differences between the forecasts can therefore also measure the importance of bidirectional coupling between the troposphere and the stratosphere in setting up the forecasting scene at initialization. If the scrambling were statistically random, and/or

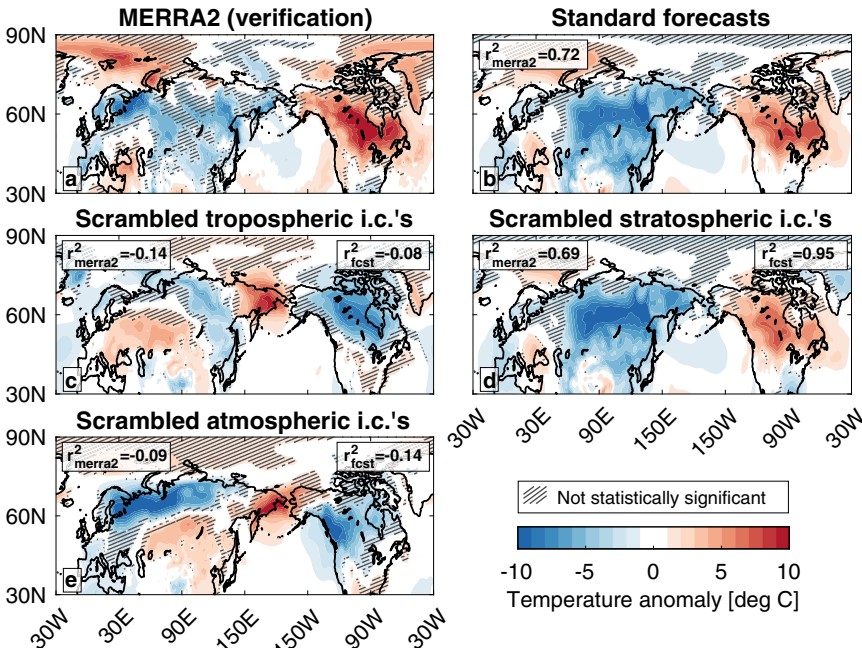

**Fig. 1 Average surface temperature anomalies in the first 2 weeks after the SSW.** January 5–18 (weeks 1–2) surface temperature anomalies from **a** MERRA2, and ensemble-mean surface temperature anomalies in the **b** standard forecast with accurate initial conditions, **c** forecast with scrambled tropospheric initial conditions, **d** forecast with scrambled stratospheric conditions, and **e** forecast with scrambled atmospheric conditions. Anomalies shaded every 1 °C. The squared anomaly correlation coefficient between each forecast and MERRA2 ($r^2_{merra2}$) and between each experimental forecast and the standard forecast ($r^2_{fcst}$) are displayed in each panel, with the correlation taken over the displayed geographic area. Negative values indicate an anticorrelation. Hatching indicates anomalies that are not statistically significantly different from the climatology at the 95% confidence level.

performed at initialization rather than during spin-up, our interpretation of these forecasts would also have to consider the influence of random variability.

**Surface weather**. Two-meter (hereafter, "surface") temperature anomalies in the two weeks following the January 5th SSW displayed the characteristic NAM response[27] that follows—and is usually attributed to—an SSW, with significant anomalous warmth over Canada, Alaska, and the Middle East, and significant anomalous cold over Europe and Asia (Fig. 1a). The standard forecast captures the pattern in MERRA2, explaining 72% of the spatial variance, although it predicts broader cold over Siberia and muted warmth over North America (Fig. 1b).

When the tropospheric initial conditions are scrambled, the forecast loses all resemblance to the standard forecast and is anticorrelated with both MERRA2 and the standard forecast (Fig. 1c). On the other hand, when the stratospheric initial conditions are scrambled and no SSW occurs, the forecast is indistinguishable from the standard forecast (Fig. 1d). Surface temperatures in the two weeks immediately following the SSW were driven entirely by the tropospheric circulation in isolation from the SSW. While peak surface impacts can take 1–3 weeks to materialize after an SSW[37,38], it is surprising that there is no detectable change in the forecast when the occurrence of the SSW is removed.

The forecast initialized with scrambled atmospheric initial conditions (Fig. 1e) predicts similar surface temperatures as the forecast with scrambled tropospheric initial conditions (Fig. 1c) over northern Siberia and North America. We can infer, then, that the ocean, sea ice, and land surfaces damped the warmth over North America and contributed to the cold over Siberia (Fig. 1a, e). As the only difference between these two forecasts is the presence of the SSW, we can also infer that the SSW neither drove

(compare Fig. 1b, d) nor damped (compare Fig. 1c,e) surface temperatures in the first two weeks after its occurrence.

The characteristic NAM surface temperature pattern evident in the initial weeks after the SSW (Fig. 1a) was still present in weeks 3–4 (Fig. 2a). Significant anomalous cold shifted into Siberia while Europe became anomalously warm. Meanwhile, the significant anomalous warmth over North America shifted to Northeastern Canada and left most of the continent with near-climatological temperatures. The standard forecast predicts a warm Northeastern Canada and cold Eurasia but misses many other features, explaining 30% of the spatial variance in MERRA2 (Fig. 2b). While this may seem low, the CESM2(WACCM6) forecast displayed elevated skill relative to the other models participating in the NOAA week 3–4 outlook (see Supplementary Fig. 2).

The forecast with scrambled tropospheric initial conditions shows some fidelity with the standard forecast over Eurasia (Fig. 2c). However, unlike MERRA2 or the standard forecast, the warmth over North America is shifted to the southeast and there is anomalous cold over Alaska. In spite of these differences, its resemblance to the standard forecast is higher than it is for weeks 1–2, explaining 31% of the spatial variance in the standard forecast. Even so, the forecast with scrambled stratospheric initial conditions (and no SSW) displays better agreement with MERRA2 and captures most of the spatial variance in the standard forecast (Fig. 2d).

While the ocean, sea ice, and land surfaces appear to have no direct contribution to either the modeled or observed surface temperatures in weeks 3–4 (Fig. 2e), over North America they force surface temperature anomalies similar to those produced by the forecast with scrambled tropospheric initial conditions. For this particular SSW, the surface impacts took over two weeks to materialize and even then did not exert a dominant influence on surface temperatures.

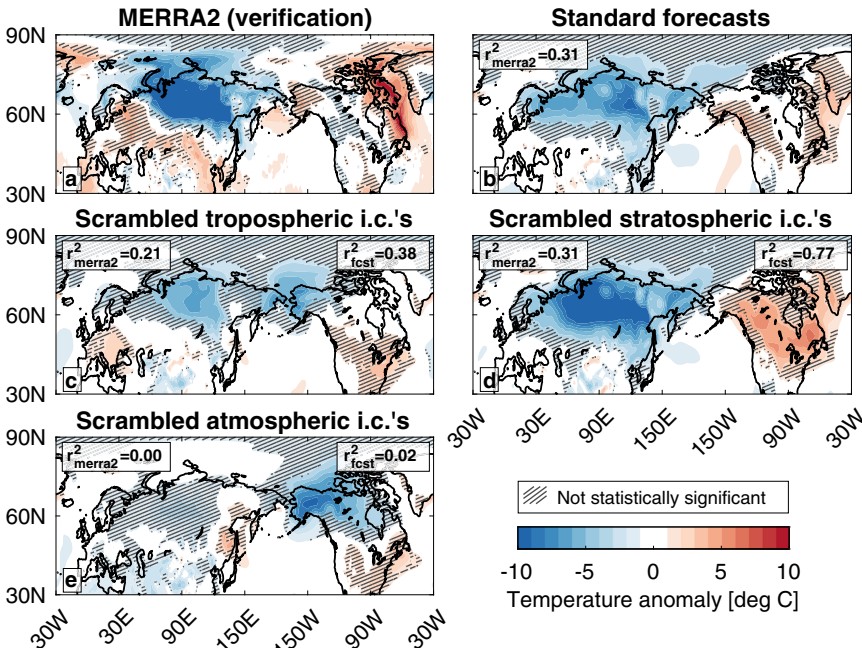

**Fig. 2 Average surface temperature anomalies 3 and 4 weeks after the SSW.** January 19–February 1 (weeks 3–4) surface temperature anomalies from (a) MERRA2, and ensemble-mean surface temperature anomalies in the **b** standard forecast with accurate initial conditions, **c** forecast with scrambled tropospheric initial conditions, **d** forecast with scrambled stratospheric conditions, and **e** forecast with scrambled atmospheric conditions. Anomalies shaded every 1 °C. The squared anomaly correlation coefficient between each forecast and MERRA2 ($r^2_{merra2}$) and between each experimental forecast and the standard forecast ($r^2_{fcst}$) are displayed in each panel, with the correlation taken over the displayed geographic area. Negative values indicate an anticorrelation. Hatching indicates anomalies that are not statistically significantly different from the climatology at the 95% confidence level.

**Stratosphere–troposphere coupling**. How could it be that the SSW had little impact on surface temperatures in the 4 weeks following the SSW, especially when surface temperatures resembled the NAM response expected after an SSW? A time series of standardized polar cap average geopotential height anomalies may provide some insight into the coupling between the troposphere and the stratosphere during this event. Geopotential height measures the altitude of a pressure level above sea level and is proportional to the average temperature of the column below.

The vertically resolved NAM index is often used to examine vertical coupling. However, as a principal component index, it describes a subset of the variability and is not always straightforward to interpret[39]. Nevertheless, polar cap average geopotential height anomalies are analogous to the NAM index[40] as both diagnose the anomalous geopotential slope between the mid-latitudes and the polar cap.

A positive geopotential height anomaly descended from the middle to the lower stratosphere in the days following the January 5, 2021 SSW, mirroring a positive geopotential height anomaly at the surface (Fig. 3a, b). In the standard forecast, both anomalies gradually weaken after January 18th, but the stratospheric anomaly persists for nearly 5 weeks (Fig. 3b). The stratospheric anomaly persisted for slightly longer in MERRA2, and a deep tropospheric anomaly emerged in mid-February (Fig. 3a).

In the forecast with scrambled tropospheric initial conditions the stratospheric geopotential height anomaly remains trapped in the middle stratosphere, suggesting that the troposphere may be an important driver of its downward propagation (Fig. 3c). It is indeed the case that when the stratospheric initial conditions are scrambled, the troposphere forces positive geopotential height anomalies in the lower stratosphere in the first 2 weeks after the SSW (Fig. 3d). In both of these forecasts, the stratospheric and tropospheric geopotential height anomalies fade faster than they

do in the standard forecast. Troposphere–stratosphere feedbacks may have helped sustain these positive geopotential height anomalies in the aftermath of the SSW (Fig. 3a, b; and see also Fig. S1—the zonal mean zonal wind at 60 N and 10 hPa recovers more rapidly in the forecast with scrambled tropospheric initial conditions). None of the forecasts reproduce the positive geopotential height anomaly in mid-February (Fig. 3a).

In the forecast with scrambled tropospheric initial conditions, a positive geopotential height anomaly emerges at the surface shortly after initialization, growing in magnitude during the first 2 weeks and linking with the stratospheric geopotential height anomaly (Fig. 3c). Rather than a confirmation of the downward influence of the SSW, it is just as consistent with the ocean, sea ice, and land forcing (Fig. 3e).

There is overall insufficient dynamical evidence to conclude that the SSW in particular, or the stratosphere in general, had a substantial impact on the surface circulation or surface temperatures in the month after the event (Figs. 1–3). The tropospheric circulation, surface forcings, and their bidirectional coupling with the stratosphere played an equal if not more important role. This is contrary to other analyses of observed geopotential heights, which have argued that this SSW likely had a strong projection onto surface weather[41,42].

**Record North American cold of February 2021**. In February of 2021, a series of winter storms swept across Canada and the United States[43] and plunged temperatures to record lows throughout the region[44]. This cold air outbreak led to an energy crisis and resulted in more than 100 deaths in Texas[45]. Media reports[46,47], forecast discussions[48], and scientific journal articles[41,42] suggested a link between this extraordinary cold and the January 5, 2021 SSW, in part based on the anomalously high

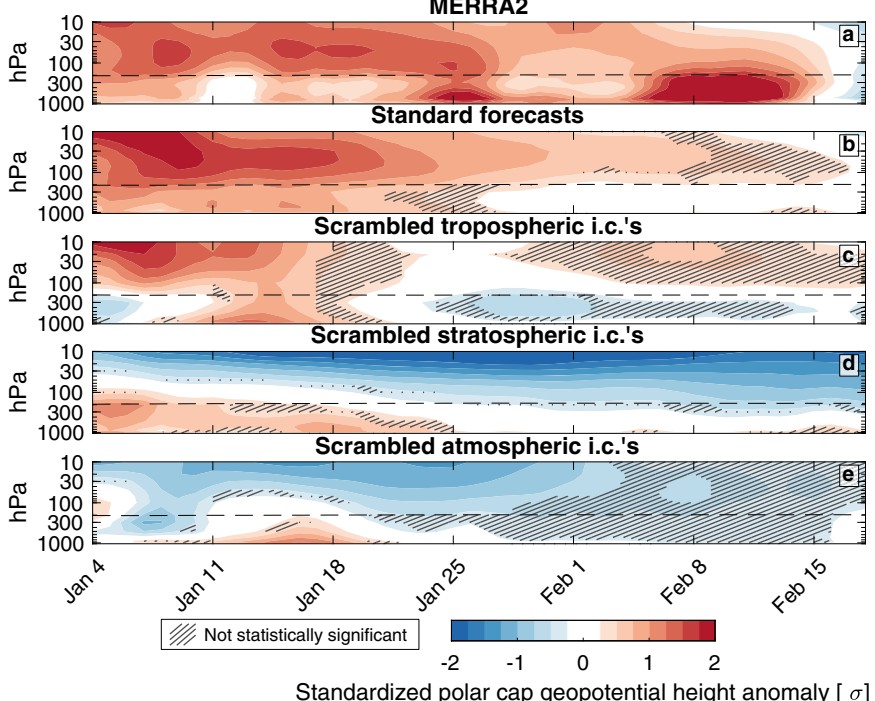

**Fig. 3 Polar cap geopotential height anomalies after the SSW.** Standardized polar cap average geopotential height anomalies in **a** the MERRA2 reanalysis, **b** the standard forecast, **c** the forecast with scrambled tropospheric initial conditions, **d** the forecast with scrambled stratospheric conditions, and **e** the forecast with scrambled atmospheric initial conditions. Positive geopotential heights indicate a warm column of air below. Anomalies are contoured every 0.25 standard deviations. The tropopause dividing the troposphere and the stratosphere is shown by the black dashed line. Hatching in **b**–**e** indicates anomalies that are not statistically significantly different from the climatology at the 95% confidence level.

polar cap average geopotential height anomalies that emerged in the troposphere (Fig. 3a).

The average temperature anomaly over Canada and the United States from February 12–18, 2021 was −5.5 °C, the coldest since the MERRA2 reanalysis initialized in 1980 (Fig. 4). At such a long forecast lead time, the standard forecast and forecast with scrambled stratospheric initial conditions initialized on January 4, 2021, do not predict extreme cold. Forecasts with scrambled tropospheric and atmospheric initial conditions predict temperature anomalies compared to the 25th percentile of the MERRA2 reanalysis climatology. Some individual forecast members predict temperature anomalies nearly as cold as observed. However, none of the forecasts are statistically significantly different from the standard forecast or the MERRA2 climatology. The extreme cold does not appear to be a deterministic outcome of the SSW.

Of course, forecast quality is poor at a 5 week lead time, especially for a series of extreme winter storms that may have been influenced by a unique confluence of events. Further, while there is an elevated risk of cold air outbreaks in the United States and Canada following an SSW, the highest risk is instead associated with polar vortex stretching[25,49–51], which generally occurs independently of an SSW[50]. To address these possibilities, we performed an additional set of forecasts initialized on February 1, 2021: a standard forecast, and a forecast with scrambled stratospheric initial conditions but accurate tropospheric initial conditions; and an additional set of forecasts initialized on February 8, 2021, less than 1 week prior to the onset of the extreme cold: a standard forecast, a forecast with scrambled tropospheric initial conditions but accurate stratospheric initial conditions, and a forecast with scrambled stratospheric initial conditions but accurate tropospheric initial conditions.

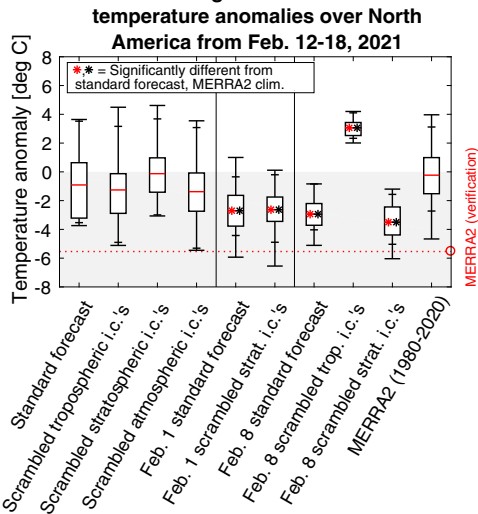

**Fig. 4 Average land surface temperature anomalies over North America during the February cold air outbreak.** Forecasted and observed land surface temperature anomalies over North America from February 12–18, 2021. Boxes indicate the 25th and 75th percentiles, large whiskers indicate the minimum and maximum, small whiskers indicate the second-lowest and second-highest values, and red lines indicate the mean. The MERRA2 climatology is shown on the far right, excluding the 2021 event. Forecasts significantly different from January 4, 2021, standard forecast, and from MERRA2 at the 95% confidence level are indicated by the red and black stars.

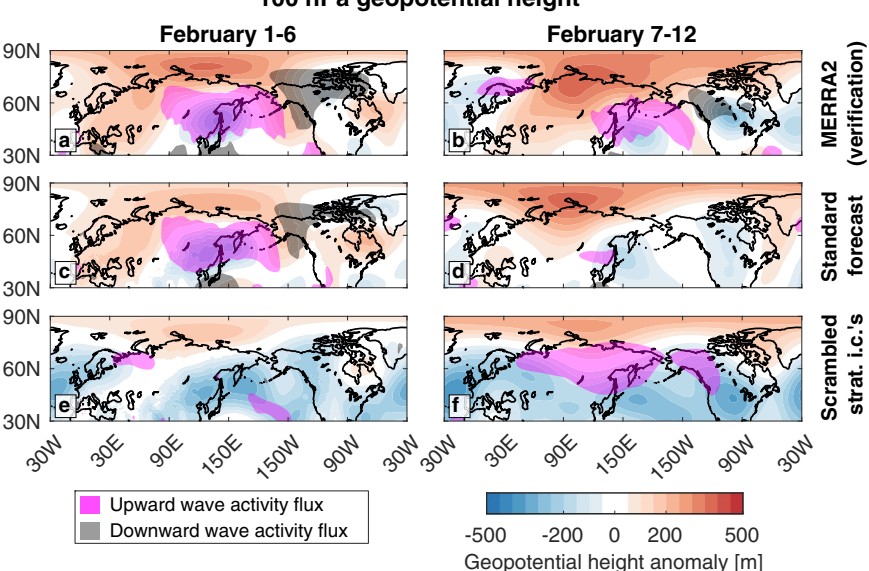

**Fig. 5 The stratospheric circulation preceding the February cold air outbreak.** 100 hPa geopotential height anomalies and vertical wave activity flux in MERRA2 **a** from February 1–6 and **b** from February 7–12, in the standard forecast initialized on February 1st **c** from February 1–6 and **d** from February 7–12, and in the forecast with scrambled tropospheric initial conditions initialized on February 1st **e** from February 1–6 and f from February 7–12. Geopotential height anomalies are shaded every 50 m, while smoothed upward wave activity flux greater than 0.2 m$^2$ s$^2$ is shaded magenta, and downward wave activity flux less than −0.2 m$^2$ s$^2$ is shaded black.

The standard forecasts and forecasts with scrambled strato-spheric initial conditions initialized on February 1st and February 8th predict temperature anomalies between −3 and −3.5 °C, comparable to the second-coldest event in the MERRA2 reanalysis (see Supplementary Fig. 3 for maps of the temperature anomalies). At least one member in each forecast predicts temperatures colder than −5 °C and all members initialized on February 8th predict anomalous cold. On the other hand, the forecast with scrambled tropospheric initial conditions initialized on February 8th predicts a temperature anomaly of 3 °C, comparable to the second-warmest event in the MERRA2 reanalysis. We emphasize that in this forecast, the tropospheric circulation at initialization is adjusted to the stratospheric circulation, even though it has drifted from its observed state. These forecasts are all significantly different from the standard forecast initialized on January 4, 2021, and from the MERRA2 reanalysis climatology.

Zonally asymmetric geopotential height anomalies reveal polar vortex stretching in the weeks preceding the cold air outbreak, with the vortex elongating over the Atlantic and narrowing over Siberia (Fig. 5a, b). Just before the event, the positive anomalies over Siberia strengthened as a strong negative anomaly emerged over North America (Fig. 5b), resembling the characteristic pattern associated with an elevated risk of cold air outbreaks[49–51].

The standard forecast reproduces the geopotential height anomalies in the days before the event (Fig. 5c, d), but does not capture the full strength of the negative anomaly over North America (Fig. 5d). On the other hand, in the forecast with scrambled stratospheric initial conditions, the troposphere forces geopotential height anomalies similar to those observed, but with a weaker positive anomaly over Siberia and a stronger negative anomaly over North America (Fig. 5e, f). In spite of this, the forecast predicts cold temperatures over North America indistinguishable from the standard forecast (Fig. 4). This raises the question of whether there is a threshold degree of vortex stretching necessary to trigger a cold air outbreak over North America, or whether polar vortex stretching is immaterial.

Reflection of planetary wave activity has been argued to be a key mechanism forcing cold air outbreaks during these polar vortex stretching events[50,51], as well as a key mechanism linking Arctic Amplification to projected changes in North American winter weather[51]. Specifically, an upward wave activity flux over Eurasia and corresponding downward wave activity flux over Alaska and Northwestern Canada (Fig. 5a, b) is hypothesized to deepen the climatological trough over North America, which could produce extreme cold weather. The standard forecast reproduces this pattern more than 1 week before the event when the trough was deepening (Fig. 5c), but not in the days immediately beforehand (Fig. 5d). However, there is no wave reflection in the forecast with scrambled stratospheric initial conditions (Fig. 5e, f). Instead, there is broad upward wave activity flux just before the event (Fig. 5f).

If wave reflection is not a necessary condition for the cold air outbreak (Fig. 4, Fig. 5e, f), why does it precede the cold air outbreak (Fig. 5a, c)? The full zonal (Fig. 6) and meridional (Fig. 7) profiles of the wave activity flux and its divergence can provide a more comprehensive perspective.

In MERRA2 and the standard forecast, the wave activity in the stratosphere between 45°N and 75°N takes a more arcing path from the midlatitudes to the pole (Fig. 6a, b) than it does in the forecast with scrambled stratospheric initial conditions (Fig. 6c). However, in all cases, all of the upward wave activity that enters the stratosphere from below remains in the stratosphere, with no wave activity penetrating the tropopause and reaching the troposphere (Fig. 6; the streamlines illustrate the instantaneous path of the wave activity). This includes the wave activity reflected downward through 100 hPa.

Granted, the most important wave dynamics for this mechanism occur in the zonal direction over the Pacific Ocean (Fig. 5), so they may be averaged out by the zonal mean perspective. In MERRA2, strong upward wave activity flux over Eurasia was reflected downward by the negative zonal winds in the middle stratosphere over the Bering Sea (Fig. 7a). However, as illustrated by the streamlines, none of this wave activity penetrated the

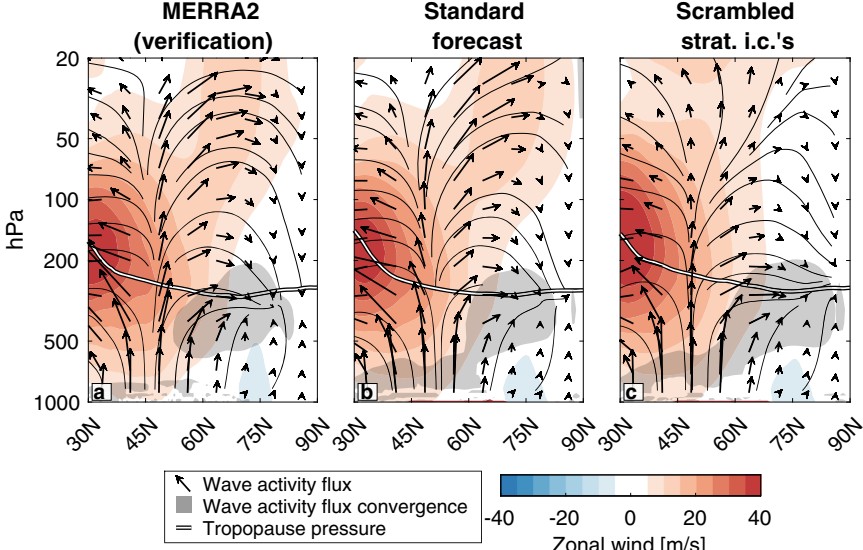

**Fig. 6 Zonal mean view of Northern Hemisphere wave dynamics preceding the February cold air outbreak.** Zonal mean (vectors) wave activity flux, (black contours) wave activity flux streamlines, (white line) tropopause, (shading) zonal wind, and (gray shading) smoothed wave activity flux divergence greater than 8 m/s/day, averaged over February 1–6, 2021, in **a** MERRA2, **b** the standard forecast initialized on February 1, 2021, and **c** the forecast with scrambled stratospheric initial conditions initialized on February 1, 2021.

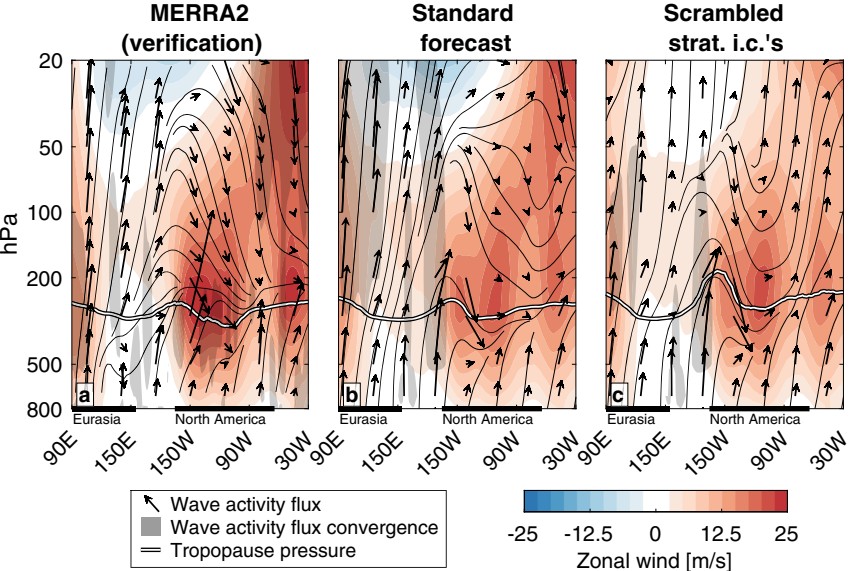

**Fig. 7 Meridional mean view of Northern Hemisphere wave dynamics preceding the February cold air outbreak.** Meridional mean (vectors) wave activity flux, (black contours) wave activity flux streamlines, (white line) tropopause, (shading) zonal wind, and (gray shading) smoothed wave activity flux divergence greater than 4 m/s/day, averaged over February 1–6, 2021 and between 45 and 75 °N, in **a** MERRA2, **b** the standard forecast initialized on February 1, 2021, and **c** the forecast with scrambled stratospheric initial conditions initialized on February 1, 2021.

tropopause. It was instead absorbed and refracted in the lower stratosphere, to the east of the low tropopause (itself a manifestation of the trough over North America). The wave activity flux convergence into the trough originated from below in the troposphere, not from above in the stratosphere.

While the standard forecast does not reproduce the full degree of downward reflection, it also shows that none of the wave activity reflected downward from the middle stratosphere reaches the troposphere (Fig. 7b). When the stratospheric initial conditions are scrambled, there is less upward wave activity over Eurasia (Fig. 7c, see also Fig. 5e), and what wave activity is

reflected downward over the Pacific is reflected back upward over North America before reaching the troposphere. In spite of this, the wave activity flux divergence in the upper troposphere/lower stratosphere relevant for the development of the trough is nearly identical between the two forecasts (Fig. 7b, c).

The cold air outbreak would have occurred whether the wave activity was reflected or not (Figs. 4 and 5), because the reflected wave activity would not have converged into the trough regardless of whether the vortex was stretched or not (Figs. 6 and 7). While vortex stretching and wave reflection may have been an important process by which the tropospheric and stratospheric

circulation evolved after the event, they do not appear to be a proximal cause of the February 2021 cold air outbreak in North America.

The wave activity flux may also be misleading if conditions were not linear[51], and this single event may have been unique in displaying these dynamical signatures without coupling to the troposphere. Alternatively, it raises the question of whether vortex stretching and wave reflection are more generally a result, rather than a cause, of the tropospheric weather regime associated with cold air outbreaks in North America. As before, this conclusion contrasts with the existing analysis and perspective of the event[52], which did not experimentally separate the influences of the tropospheric and stratospheric circulations.

## Discussion

In summary, we do not find evidence to suggest that the January 5, 2021 SSW itself, nor the polar vortex stretching and wave reflection in its aftermath, caused or otherwise influenced the record-breaking cold in North America in February of 2021, at least when using the CESM2(WACCM6) Earth system prediction framework. If anything, we find evidence that without coupling with the correct tropospheric variability, the stratosphere may have otherwise forced exceptional warmth over North America during the same period. We support these claims through careful experimentation on the resolved circulation—in particular, by preventing the wave reflection mechanism from occurring by uninitializing, or "scrambling", the stratospheric initial conditions in an ensemble forecast.

Additionally, our experimental forecasts show that the SSW that occurred on January 5, 2021, had a minor impact on surface temperatures, explaining none of the spatial variances in the first and second weeks (Fig. 1c) and at most one-third of the spatial variance in the third and fourth weeks (Fig. 2c) after the event. Surface temperature variability during these four weeks was primarily governed by the influence of persistent tropospheric weather in isolation from the SSW (Fig. 1d; Fig. 2d). While this analysis focused on a single SSW, there are previous events during which a persistent tropospheric circulation present before an SSW may have influenced surface temperature anomalies in the weeks afterward[53].

The common interpretation of the pattern of positive geopotential height anomalies following an SSW is a downward propagation of the negative phase of the NAM that periodically "drips" into the troposphere and impacts surface weather[54] (Fig. 3a). During this event the tropospheric circulation induced the lower stratospheric geopotential height anomalies in the aftermath of the SSW, effectively drawing the SSW signal downward[55] (Fig. 3c).

Rather than downward propagation, these dynamics bear more resemblance to a feedback process. If wave coupling feedbacks between the lower stratospheric and surface geopotential height anomalies increased their persistence following the SSW[56] (Fig. 3a), the SSW may have been relevant for surface weather insofar as it reinforced and sustained the tropospheric-forced lower stratospheric geopotential height anomalies[56–61].

In apparent contradiction to these results, previous work has demonstrated that constraining the stratospheric (but not tropospheric) circulation to observations after SSWs demonstrates that the negative NAM in the stratosphere on average leads to a negative NAM in the troposphere[60]. However, such an experimental design effectively bakes in stratosphere-troposphere feedbacks and any tropospheric driving of the downward progression of geopotential height anomalies (Fig. 3d). The ambiguity inherent to a "perfect stratosphere" simulation makes it difficult to determine whether it is the SSW, or the tropospheric

circulation at SSW onset, or stratosphere–troposphere feedbacks that lead to the negative NAM at the surface.

Here, we have shown that scrambling the initial conditions of different aspects of the atmosphere can be valuable for assessing the physics of individual events in real-time. It also complements existing statistical techniques that attempt to isolate causal relationships[62] and counter the tendency to over-interpret causality[63]. Initial condition scrambling can be readily extended to other extreme events in the Earth system to improve estimates of forecast uncertainty, guide model development priorities, and provide greater clarity for attribution studies.

## Methods

**Anomaly calculations.** Daily average surface (2-m) temperature and geopotential height anomalies are calculated as deviations from the 1999–2019 daily average for both MERRA2 and CESM2(WACCM6) and are lead-dependent for CESM2(WACCM6). As CESM2(WACCM6) hindcasts are only generated every Monday, a 120-day triangular smoothing is applied to the raw climatology to ensure an adequate representation of the seasonal cycle. The same smoothing is applied to the MERRA2 climatology so that all comparisons are fair.

**Standardization.** Geopotential height anomalies are standardized by subtracting the daily climatology and dividing by the daily standard deviation and are lead-dependent for CESM2(WACCM6).

**The squared anomaly correlation coefficient.** The squared anomaly correlation coefficient is the squared correlation coefficient between two sets of surface temperature anomalies weighted by the cosine of latitude. We square the correlation and multiply it by the sign of the correlation so that it both (1) communicates the fraction of spatial variance shared and (2) identifies whether the two sets are correlated or anti-correlated.

**Tropopause calculation.** The tropopause is calculated as the lowest level where the lapse rate, $dT/dz$, drops below 2 K/km and stays below 2 K/km between the level and all levels up to 2 km above. The daily polar cap average tropopause pressure is displayed in Fig. 3.

**Area averages.** The polar cap average is the 60–90°N average weighted by the cosine of latitude, while the mid-latitude average is the 45–75°N average weighted by the cosine of the latitude.

**North America land average.** The North America land average is taken over a latitude-longitude rectangle defined by latitudes 25°N and 70°N and longitudes 70°W and 135°W. Beginning from the southeast corner and moving clockwise, the rectangle runs from due east of The Bahamas; to due north of the Hawaiian Islands; to due northwest of the northwestern coast of Alaska; to Baffin Island. The average is cosine-weighted and only taken over the land. It is centered on the United States of America and Canada, which experienced extreme and widespread cold surface temperature anomalies in mid-February 2021. The results of Fig. 4 are relatively insensitive to the choice of bounding latitudes and longitudes.

**February 12-18, 2021 event.** North America land average temperature anomalies are averaged from February 12–18, 2021. Averages are taken from February 12–18 every year from 1980 through 2020 to construct the MERRA2 climatology.

**Wave activity flux.** The quasi-geostrophic zonal, meridional, and vertical wave activity flux vector for stationary waves is given by[64]

$$\{F^\lambda, F^\phi, F^z\} = p\cos(\phi)\left\{ v'^2 - \frac{1}{2\Omega a\sin(2\phi)}\frac{\partial(v'\Phi')}{\partial\lambda}, -u'v' \right.$$
$$\left. + \frac{1}{2\Omega a\sin(2\phi)}\frac{\partial(u'\Phi')}{\partial\lambda}, \frac{2\Omega\sin(\phi)}{\frac{\partial\hat{T}}{\partial z}+\frac{k\hat{T}}{H}}\left(v'T' - \frac{1}{2\Omega a\sin(2\phi)}\frac{\partial(T'\Phi')}{\partial\lambda}\right) \right\} \quad (1)$$

where $p$ is the pressure, $a$ is the radius of the earth, $\phi$ is latitude, $u$ is the zonal wind, $v$ is the meridional wind, $\Omega$ is the rotation rate of the earth, $\Phi$ is the geopotential, $\lambda$ is longitude, $z$ is the log-pressure height, $k$ is the ratio of the specific gas constant of dry air to the specific heat of dry air, $T$ is the temperature, hats indicate a domain average, $H$ is the log-pressure scale height, and primes denote deviations from the zonal mean. The log-pressure height is given by $z = -H\ln(p/p_0)$, where $p_0$ is the reference pressure.

The zonal mean quasi-geostrophic wave activity flux is given by[65]

$$\{F^\phi, F^p\} = \rho_0\, a\cos(\phi)\left\{ -[u'v'], 2\Omega\sin(\phi)\frac{[v'\theta']}{\frac{\partial\theta}{\partial z}} \right\} \quad (2)$$

where $\rho_0$ is the log-pressure density, $\theta$ is the potential temperature, and brackets denote the zonal mean. The log-pressure density is given by $\rho_0 = p/Hg$.

We set the log-pressure scale height to 7 km and the reference pressure to 1000 hPa. For Earth, $\Omega = 7.3 \times 10^{-5}\,\mathrm{s}^{-1}$, $a = 6.4 \times 10^6\,\mathrm{m}$, $g = 9.81\,\mathrm{m/s}^2$, and $k = 0.29$.

In all plots, wave activity flux vectors are scaled so that their direction and magnitude are physically sound for the given aspect ratio and domain[65]. The scaled horizontal and vertical components are

$$\hat{F}^\phi = F^\phi \cdot \frac{X/Y}{\phi_1 - \phi_0} \tag{3}$$

$$\hat{F}^\lambda = F^\lambda \cdot \frac{X/Y}{\lambda_1 - \lambda_0} \tag{4}$$

$$\hat{F}^z = F^z \cdot \frac{1}{z_1 - z_0} \tag{5}$$

where $X$ and $Y$ are the lengths of the horizontal and vertical axes, and subscripts indicate the maximum and minimum axis values.

Streamlines are drawn parallel to this vector field to indicate the instantaneous propagation pathway for wave activity. This provides a more objective interpretation of the wave activity flux than flux vectors alone.

**CESM2(WACCM6).** CESM2 is an Earth system model with prognostic atmosphere, ocean, land, sea ice, land ice, river, and wave components[34]. WACCM6 is the configuration of the atmosphere with 70 vertical levels from the surface to approximately 140 km in the lower thermosphere[35]. CESM2(WACCM6) forecasts use the default configuration of the model, which couples a finite-volume dynamical core with a 1.25° longitude by 0.9° latitude rectilinear grid to convection, turbulence, microphysics, gravity wave drag, and aerosol physics parameterizations. CESM2(WACCM6) includes prognostic aerosols and prognostic troposphere-stratosphere-mesosphere-lower thermosphere chemistry with 228 advected chemical species.

The ocean is prognostically modeled with the Parallel Ocean Program[66] Version 2, while sea ice is modeled with the Community Ice CodE[67] Version 5.1.2. Both models use a uniform 1.25° longitude and variable 1.0–0.27° latitude rectilinear grid. The land surface, including soil moisture and vegetation, is modeled with the Community Land Model[68] Version 5, while river runoff is modeled by the Model for Scale Adaptive River Transport[69]. Ice sheets on land are prescribed with the Community Ice Sheet Model[70] Version 2.1.

Initial conditions are generated from a specified dynamics (SD) version of CESM2(WACCM6) in which the horizontal winds and temperatures are nudged to their values in the National Aeronautics and Space Administration (NASA) Forward Processing for Instrument Teams (FP-IT) near-real-time reanalysis at a relaxation timescale of 1 h. This short timescale ensures that the observed state of the atmosphere, including chemistry and aerosols, is represented as accurately as possible in CESM2(WACCM6) at initialization. Ocean and sea ice initial conditions are drawn from this simulation, as well. Initial conditions for the land model are generated from a stand-alone land simulation forced by the National Centers for Environmental Prediction Climate Forecast System[71] Version 2. The random field perturbation method[72] is used to generate an ensemble of forecasts from the CESM2(WACCM6) SD simulation every Monday.

For forecasts with scrambled tropospheric initial conditions, initialized on both January 4 and February 8, 2021, a new CESM2(WACCM6) SD run is branched starting from December 21, 2020, with nudging tapered from a 1 h timescale at 20 km to an infinite timescale at 15 km (no nudging). For forecasts with scrambled stratospheric initial conditions, another new CESM2(WACCM6) SD run is branched starting from November 16, 2020, with nudging tapered from a 1 h timescale at 9 km to an infinite timescale at 12 km (no nudging). This earlier date was chosen as initializing on December 21, 2020, resulted in an SSW in the SD simulation on January 1, 2021. Initial conditions for the atmosphere are seeded from these SD simulations. For the forecasts with scrambled atmospheric initial conditions, initial conditions for the atmosphere are drawn from January 2, 2017, as 2017 was a year with no major SSW.

The zonal wind at 60°N and 10 hPa from all forecasts are displayed in Fig. S1 in the Supplementary Information. Twenty-one of 21 standard forecasts, 12 of 21 forecasts with scrambled tropospheric initial conditions, 0 of 21 forecasts with scrambled stratospheric initial conditions, and 1 of 21 forecasts with scrambled atmospheric initial conditions produce an SSW.

**Verification.** We use the NASA Modern-Era Retrospective Reanalysis Version 2[73] assimilation product as our verification dataset, which spans 1980 to the present. MERRA2 is provided on a 0.5° latitude by 0.66° longitude grid with 71 vertical levels.

## Data availability
MERRA2 and FP-IT is provided by the Global Modeling and Assimilation Office at the NASA Goddard Space Flight Center. MERRA2 can be accessed from the NASA Goddard Earth Sciences (GES) Data and Information Services Center (DISC) at https://

disc.gsfc.nasa.gov/datasets?project=MERRA-2 (registration may be required). For access to FP-IT, contact geos5-nrt@lists.nasa.gov. All raw CESM2(WACCM6) output necessary to replicate these results is available at https://doi.org/10.5281/zenodo.5639805.

## Code availability
All scripts used in the analysis are permanently archived at https://doi.org/10.5281/zenodo.5645655.

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

## Acknowledgements
The CESM project is supported primarily by the National Science Foundation (NSF). This work was supported by NOAA's Weather Program Office/Climate Test Bed program and by the National Center for Atmospheric Research, which is a major facility sponsored by the National Science Foundation under Cooperative Agreement 1852977. Portions of this study were supported by the Regional and Global Model Analysis (RGMA) component of the Earth and Environmental System Modeling Program of the U.S. Department of Energy's Office of Biological and Environmental Research (BER) via the National Science Foundation IA 1844590. Computing and data storage resources, including the Cheyenne supercomputer (doi:10.5065/D6RX99HX), were provided by the Computational and Information Systems Laboratory (CISL) at NCAR.

## Author contributions
J.H.R., J.E. and A.A.G. developed and manage the CESM2(WACCM6) Earth system prediction framework. N.A.D. downloaded and processed the verification dataset. E.L.J. collected and processed model output from other modeling centers for Fig. S2. J.H.R., A.A.G. and N.A.D. conceived of the project and the experimental design. J.H.R. and A.A.G. ran the forecasts and N.A.D. analyzed the forecasts, made the figures, and wrote the paper. All authors contributed to revising the paper.

## Competing interests
The authors declare no competing interests.
