## [Peer Review File · Nature Communications]

Review of *Davis et al.* “Limited surface impacts of the January 2021 sudden stratospheric warming”

NCOMMS-21-14662

This study analyses stratosphere-troposphere coupling following the most recent major SSW using a set of experiments involving subseasonal predictions from the CESM2-WACCM6 model. The experiments aim to isolate the role of the stratospheric warming event in the tropospheric evolution in the subsequent weeks to months. The authors conclude that the surface pattern would have evolved similarly without the SSW. A further piece of analysis focuses on the high-impact Texas cold wave during mid-February, which was widely attributed at the time to the effects of the SSW. The study finds little evidence to support this claim.

The article is very clearly written, and easy to follow. The analysis focuses on a recent weather event which gained widespread media attention in both Eurasia and North America, so the results of the study would be of wider interest and important in many subsequent studies.

However, I do not think that the figures presented, or the corresponding analysis, are sufficient to support some of the strong claims made by the authors. Further, I believe some of the assertions conflate *predictability* with *causality*. Having said that, I do think that the study can be a valuable contribution but requires major revision. Specific comments below.

Major comments:

L108-109/Figure 1: is there a particular reason why a simple pattern/anomaly correlation coefficient is not used here (which is more standard)? The correlation coefficients themselves are quite high; squaring them gives an impression of a lower correlation. Please state/confirm that cosine area weighting is included.

Figure 1 & 2: There are seemingly no oceanic/Arctic NOAA CPC observations in panel (a), but there are data for the forecasts. Not only is this a discontinuity, but makes me question how the correlations were calculated and whether they are affected by this? Please check/confirm. If these are not available, then using a modern reanalysis might be required.

Figure 1 & 2: While I think it is worthwhile showing T2m somewhere in the study, I would be more interested in seeing the Z500/Z1000 or MSLP anomalies, to understand the dynamics at play – since T2m is noisy and perhaps a more challenging variable to predict. Rather than discussing whether the temperatures look like what would be expected from a NAM pattern (e.g., L106, L135), why not just show the flow anomalies themselves? Different dynamical patterns can produce similar temperature anomalies, so I strongly suggest that Fig 1 & 2 are moved to supporting information and replaced in the main text with a geopotential height anomaly or MSLP map.

Figure 3: My primary concern here is that there is no comparison with the observed NAM (I would like the authors to add this to the paper in some form). The Standard forecasts do not capture the full structure of the evolution beyond late Jan (which is to be expected) when there was a re-intensification of the stratospheric anomalies and secondary deceleration of U10-60 (also not captured in the U forecasts by looking at Fig. S1). So, surely analysis of anything beyond this relates to *predictability*, rather than *causality*? This is interesting, but to assess

causality I would argue the approach requires relaxing the stratosphere to observations and then assessing the tropospheric evolution.

Further, it is very hard to see the weak negative NAM at the surface. And in the real world, this negative NAM structure extended through the depth of the troposphere, so it is not as though the scrambled stratosphere runs produced something which strongly resembled what occurred in 2021. Aside from this, to fully illustrate the point, I would like to see a comparison ensemble timeseries of the 1000 hPa NAM, which shows *how much more likely/intense* the negative NAM was in the different forecasts, versus the observed NAM. These are otherwise very weak signals of an ensemble mean which might be very sensitive to a few ensemble members. I do not think this figure is enough to fully claim that a negative surface NAM would have occurred without the SSW.

L237-241: This is where I am particularly concerned at the distinction between causality and predictability. The Texas cold wave was an extreme synoptic event taking place on a timescale of only a few days. I do not think this would be predictable at all at this lead-time, given it might require a perfect interaction of stratospheric forcing (or otherwise) with other subseasonal phenomena (the MJO was active during this time, for example). As it stands, to me this analysis just says that this model cannot predict an extreme temperature event 5 weeks ahead of time... which is not surprising. If the standard forecasts predicted the cold wave, and those with the scrambled stratospheric ICs also did, then I would agree that the stratospheric forcing was not important. But since neither predicted the cold wave, then how can we conclude the stratosphere was not important? A better approach might be to find the lead-time at which the standard forecast was able to predict the event, and then perform these experiments. As it stands, the conclusion I take from this is “subseasonal prediction of the Texas cold wave was not increased by the occurrence of the SSW”. However, I do agree with the authors that many were far too quick to attribute causality at the time.

Looking at Fig. 3a, there is no substantial signal of a negative NAM through the depth of the troposphere around the time of the Texas cold wave, as was observed – so it is not surprising to see an absence of the surface thermal signal.

Minor comments:

L14: are Arctic SSWs ‘rare’? (same comment for L53) Once every other winter is relatively common, to my mind, in comparison with e.g., Antarctic SSWs. Further, the text here should specifically state Arctic SSWs.

L15-17: I found this sentence hard to understand. Additionally, there seems to be some contradiction between the statements for “North America” and “Eastern United States”, while I find the statement that there is ‘stormy’ weather in Eurasia perplexing... at least in Europe, it is NAO+/strong polar vortex conditions associated with ‘stormy’ weather.

L31: Is it the database which is standardized, or the forecast? I suggest “standardized databases of subseasonal forecasts.”

L44: “could”... perhaps “can”?

L48: Suggest “associated with *breaking* planetary waves”

L49: Suggest changing “producing” to “known as”. It might be worth mentioning something about what constitutes a *major* event.

L53: It is alluded to but maybe it could be more clearly stated that the wave activity need not be anomalous, e.g. Birner and Albers 2017 SOLA, de la Camara et al. 2018 J Clim

L62-63: I am not sure I understand “storminess increases the variability”.

L65: What is the “scientific novelty”? SSWs are not novel.

L79: “often”... is “usually” better?

L84: I think this sentence makes sense without “the meteorology in”

L115: The absence of a notable impact within 2 weeks of the SSW occurrence seems within expectation, since it is broadly dominated by the timescale for the tropospheric NAM (e.g., Simpson et al. 2011 <https://doi.org/10.1029/2011GL049304>) and usually not long enough after the SSW for a typical downward impact to be noted (e.g., Hall et al. 2020 JGR <https://doi.org/10.1029/2020JD033881>). Some of the preconditioning of the vortex before the SSW itself may have played a role though (lower-mid stratospheric NAM was negative before 5 January) which is perhaps worth noting.

L170: Reference to ‘anomalies’ is needed, i.e., “polar-cap average geopotential height *anomalies* are similar to the NAM index, as both diagnose the *anomalous* geopotential slope”.

L185: It is very interesting that the troposphere may have enhanced the downward propagation. Analysis of the wave activity/eddy heat flux may show this was important – the eddy heat flux did remain high in the lower-stratosphere following the SSW onset. But probably beyond the scope of this study.

L249-250: Yes, but there are many ways in which the atmosphere can produce cold conditions in Texas – not necessarily in a flow evolution which resembles what happened in 2021.

L266: Please state how anomalies were standardized.

L288: “Insensitive” to what extent? As it is a small domain, surely it becomes sensitive at some point?

L326: There was a *minor* SSW involving a significant vortex deceleration during late Jan 2017, but yes – there was no *major* event, and the vortex was strong on the date noted.

Figure 3: the very thin negative anomalies near the surface in the first few days of the Standard forecast are dubious to me.

Review of “Limited surface impacts of the January 2021 sudden stratospheric warming” by Richter et al.

The study compares the contributions of the stratosphere versus the troposphere and surface boundary conditions to extratropical surface weather following the Stratospheric Sudden Warming (SSW) of January 2021. The causal role of each component is assessed using targeted experiments in a dynamical prediction model, where the initial conditions of a given component are set to observations while the others are “scrambled”. The results argue for a limited influence of the SSW on the surface conditions that followed, particularly in Texas which experienced severely cold weather.

The approach of using a dynamical prediction model to assess the causal impact of a SSW on surface weather is technically sound. However, I found two issues with the study which may impact its publication in this journal. The first is that I’m unsure whether there is sufficiently strong evidence to support some of the claims, but this could potentially be addressed with some additional analysis. The second issue, following the stated aims and scope of this journal, is that I’m unsure this study represents a sufficiently ‘important scientific advance of significant interest to specialists’. This could potentially be overcome if the authors frame their results a bit differently. These issues are explained in further detail below to help the authors improve their manuscript and help the editor reach a decision.

General comments:

1. I’m unsure whether there is sufficiently strong evidence to support some of the claims made in this study. It seems to me that in order to use the scrambled experiments to assess causality of the event, the standard/control forecast must first be able to capture the event itself. Otherwise, the scrambled experiments are isolating the causal influence of a different event. Figure 1b shows that the standard forecast captures an important fraction (50%) of the spatial pattern of temperature anomalies during the two weeks following the event. However, the fraction becomes progressively worse later in the forecast (Fig. 2b) to the point where the standard forecast does not capture the same event as observations (Fig. 4). This issue could potentially be addressed by repeating the same analyses except using forecasts initialised closer to the dates of the anomalies shown. This is because the goal of this study is not to demonstrate the ability of the model to forecast the observed surface weather at long-lead times but rather to establish the causal role of the stratosphere versus the other components.
2. As it stands, I’m unsure whether this study represents a sufficiently ‘important scientific advance of significant interest to specialists’. There have been several other studies in specialist journals which explored why some SSWs couple with the troposphere while others do not, sometimes called ‘propagating’ versus ‘non-propagating’ SSWs [Gerber et al., 2009, Jucker, 2016, Karpechko et al., 2017, Nakagawa and Yamazaki, 2006, Rao et al., 2020, White et al., 2019]. To be fair, most of these studies assume that similar signed NAM anomalies in the troposphere and stratosphere following SSWs imply a causal influence from the stratosphere which is not necessarily the case as discussed here. However, Hitchcock and Simpson [2014] isolated the causal role of the stratosphere using stratospheric nudging experiments in a state-of-the-art climate model and showed

that SSWs tend to shift the distribution of tropospheric weather favouring some conditions over others (see their Fig. 6). Thus, the results presented here are not really surprising because we expect that a large fraction of the stratospheric influence on the troposphere will be masked by internal tropospheric variability [Gerber et al., 2009]. Finally, the fact that this specific SSW event received a lot of media attention strikes me as insufficient motivation for specialists and more interesting to the general public.

I think the authors could frame their results differently, and thus increase the impact and novelty of their study, by arguing for the feasibility of their 'scrambling' approach for real-time attribution of events using operational forecasts models. To my knowledge, the scrambling approach has only been used to assess the causal role of the stratosphere in an idealized model [Gerber et al., 2009] and the results presented here could potentially be the first to demonstrate its use in a realistic setting. Since there is currently a lot of interest in 'windows of opportunity' during S2S forecasts, particularly from stratospheric sources, encouraging prediction centers to implement this type of analysis could be useful for understanding and improving forecast skill. The scrambling approach could be complementary to and easier to implement than the nudging approach which is currently being explored as part of the SNAPSI effort.

Specific comments:

1. L50-53: A good reference here would also be Dunn-Sigouin and Shaw [2020].
2. L56-63: I think the authors should also refer to the studies cited above about propagating versus non-propagating SSWs.
3. L131-141, Fig. S2 and Table S1: considering major comment 1 above, I don't think showing how poorly other models forecast the anomalies at long lead-times helps to support the results.
4. L183-185: a relevant citation here would be Plumb and Semeniuk [2003]
5. L195-197: I think this could be rephrased for clarity. The authors could also mention explicitly that their results are opposite those published previously about this event.
6. L207 and elsewhere: I suggest replacing 'variance' with 'spatial variance' for clarity.
7. Figs 1,2 and 3: I think it would be helpful to include a significance measure of the anomalies shown. The forecast could have a similar mean to observations but could also be highly uncertain and thus have low significance.
8. Fig. 3: Could the authors include a panel with observations for comparison as in the other figures?
9. Analysis: Have the authors considered looking at other weather relevant variables like precipitation? Would the results be similar ?
10. Analysis: How did the authors calculate the squared spatial correlation coefficient ? Is this similar to the anomaly correlation coefficient (ACC, e.g., equation 2 in Domeisen et al. [2020]). Have the authors looked at other commonly used skill measures of their forecasts ? Would the results be consistent ?

References

- Domeisen, D., et al., 2020: The role of the stratosphere in subseasonal to seasonal prediction part i: Predictability of the stratosphere. *Journal of Geophysical Research: Atmospheres*.
- Dunn-Sigouin, E. and T. Shaw, 2020: Dynamics of anomalous stratospheric eddy heat flux events in an idealized model. *Journal of the Atmospheric Sciences*, **77 (6)**, 2187–2202.
- Gerber, E., C. Orbe, and L. M. Polvani, 2009: Stratospheric influence on the tropospheric circulation revealed by idealized ensemble forecasts. *Geophysical Research Letters*, **36 (24)**.
- Hitchcock, P. and I. R. Simpson, 2014: The downward influence of stratospheric sudden warmings. *Journal of the Atmospheric Sciences*, **71 (10)**, 3856–3876.
- Jucker, M., 2016: Are sudden stratospheric warmings generic? insights from an idealized gcm. *Journal of the Atmospheric Sciences*, **73 (12)**, 5061–5080.
- Karpechko, A. Y., P. Hitchcock, D. H. Peters, and A. Schneidereit, 2017: Predictability of downward propagation of major sudden stratospheric warmings. *Quarterly Journal of the Royal Meteorological Society*, **143 (704)**, 1459–1470.
- Nakagawa, K. I. and K. Yamazaki, 2006: What kind of stratospheric sudden warming propagates to the troposphere? *Geophysical research letters*, **33 (4)**.
- Plumb, R. A. and K. Semeniuk, 2003: Downward migration of extratropical zonal wind anomalies. *Journal of Geophysical Research: Atmospheres*, **108 (D7)**.
- Rao, J., C. I. Garfinkel, and I. P. White, 2020: Predicting the downward and surface influence of the february 2018 and january 2019 sudden stratospheric warming events in subseasonal to seasonal (s2s) models. *Journal of Geophysical Research: Atmospheres*, **125 (2)**.
- White, I., C. I. Garfinkel, E. P. Gerber, M. Jucker, V. Aquila, and L. D. Oman, 2019: The downward influence of sudden stratospheric warmings: Association with tropospheric precursors. *Journal of Climate*, **32 (1)**, 85–108.

Response to Reviewers

We thank the two anonymous reviewers for their feedback. We'd like to summarize major changes to the manuscript before addressing your individual comments.

1. Addition of a set of forecasts initialized on February 8th to Figure 4, initialized less than one week prior to the extreme cold event. The forecasts include a standard forecast, which reliably captures the extreme cold, and a "perfect stratosphere" forecast with scrambled tropospheric conditions which predicts exceptional warmth. We believe this more reliably demonstrates that the January 5th, 2021 SSW did not drive the extreme cold over North America in February. We also chose to expand the averaging region to encompass North America. We agree with the reviewers that it is difficult to infer causality when examining weather over a small region. Even when averaged at this continental scale, it was the coldest event on record since MERRA2 began in 1980.
2. Additional discussion of the merits of the initial condition approach versus the existing "perfect stratosphere" simulations of Hitchcock and Simpson (2014) and proposed by SPARC SNAPSI. We present an argument that these "perfect stratosphere" simulations are more difficult to interpret than our initial condition approach, given the importance of upward coupling from the troposphere for this event (Fig. 3).
3. Addition of MERRA2 to Figures 1-4, and removal of NOAA CPC Global Temperature. We originally used the NOAA CPC Global Temperature dataset as it is the observational data product used by NOAA SubX for verification purposes, but we recognize the interest in temperatures over the oceans, and a verification dataset for geopotential height. The impacts of this on our results are (1) a greater forecast skill for CESM2(WACCM6) forecasts, as it performs especially well over the ocean basins, and (2) a verification for geopotential height, which provides some basis for our discussion of the geopotential forecasts.
4. Reordered Figure 4 and its discussion to follow Figure 3 and its discussion; combined the "Conclusions" and "Discussion" sections to be more succinct.

Please see the tracked-changes version of the manuscript to view how your comments and suggested revisions have been implemented.

Thank you for your time and effort, we appreciate it,
Nick Davis & coauthors

Review of Davis et al. “Limited surface impacts of the January 2021 sudden stratospheric warming”

NCOMMS-21-14662

This study analyses stratosphere-troposphere coupling following the most recent major SSW using a set of experiments involving subseasonal predictions from the CESM2-WACCM6 model. The experiments aim to isolate the role of the stratospheric warming event in the tropospheric evolution in the subsequent weeks to months. The authors conclude that the surface pattern would have evolved similarly without the SSW. A further piece of analysis focuses on the high-impact Texas cold wave during mid-February, which was widely attributed at the time to the effects of the SSW. The study finds little evidence to support this claim. The article is very clearly written, and easy to follow. The analysis focuses on a recent weather event which gained widespread media attention in both Eurasia and North America, so the results of the study would be of wider interest and important in many subsequent studies. However, I do not think that the figures presented, or the corresponding analysis, are sufficient to support some of the strong claims made by the authors. Further, I believe some of the assertions conflate predictability with causality. Having said that, I do think that the study can be a valuable contribution but requires major revision. Specific comments below.

Major comments:

L108-109/Figure 1: is there a particular reason why a simple pattern/anomaly correlation coefficient is not used here (which is more standard)? The correlation coefficients themselves are quite high; squaring them gives an impression of a lower correlation. Please state/confirm that cosine area weighting is included.

We have updated the name of this diagnostic to “squared anomaly correlation coefficient”, and now clarify in the “Methods” section that this is the anomaly correlation coefficient squared; it is multiplied by the sign of the anomaly correlation coefficient; and that it is cosine-weighted. We square it because the square of the correlation coefficient quantifies the fraction of variance shared between two records, which is a useful physical interpretation.

Figure 1 & 2: There are seemingly no oceanic/Arctic NOAA CPC observations in panel (a), but there are data for the forecasts. Not only is this a discontinuity, but makes me question how the correlations were calculated and whether they are affected by this? Please check/confirm. If these are not available, then using a modern reanalysis might be required.

There are no oceanic NOAA CPC observations, as it is a land-only dataset. The correlations were performed correctly, with oceanic grid points omitted. We do agree it would be worth showing the oceanic temperature anomalies, so we replaced NOAA CPC with MERRA2. In general the correlations are a bit higher, both in absolute terms and relative to other models.

Figure 1 & 2: While I think it is worthwhile showing T2m somewhere in the study, I would be more interested in seeing the Z500/Z1000 or MSLP anomalies, to understand the dynamics at play – since T2m is noisy and perhaps a more challenging variable to predict. Rather than discussing whether the temperatures look like what would be expected from a NAM pattern (e.g., L106, L135), why not just show the flow anomalies themselves? Different dynamical patterns can produce similar temperature anomalies, so I strongly suggest that Fig 1 & 2 are moved to supporting information and replaced in the main text with a geopotential height anomaly or MSLP map.

We focused on temperature because it is one of the most important forecasted variables. It impacts human activity, industry (including electrical utilities), and ecology. Cold-core lows are characteristic of midlatitude Rossby wave dynamics, so we wouldn't expect the sea-level pressure field to be more informative or reveal a substantially different story. We've added the +/- 3, 6, and 9 hPa sea-level pressure anomaly contours to the temperature plot shown below to illustrate that it does not add any useful discussion - the standard forecast and forecast with scrambled stratospheric initial conditions look closer to MERRA2 than the forecasts with scrambled tropospheric and atmospheric initial conditions, as is the case for temperature. We think our focus on zonal mean geopotential height is as detailed as necessary to illustrate the circulation.

Figure 3: My primary concern here is that there is no comparison with the observed NAM (I would like the authors to add this to the paper in some form). The Standard forecasts do not

capture the full structure of the evolution beyond late Jan (which is to be expected) when there was a re-intensification of the stratospheric anomalies and secondary deceleration of U10-60 (also not captured in the U forecasts by looking at Fig. S1). So, surely analysis of anything beyond this relates to predictability, rather than causality? This is interesting, but to assess causality I would argue the approach requires relaxing the stratosphere to observations and then assessing the tropospheric evolution.

Further, it is very hard to see the weak negative NAM at the surface. And in the real world, this negative NAM structure extended through the depth of the troposphere, so it is not as though the scrambled stratosphere runs produced something which strongly resembled what occurred in 2021. Aside from this, to fully illustrate the point, I would like to see a comparison ensemble timeseries of the 1000 hPa NAM, which shows *how much more likely/intense* the negative NAM was in the different forecasts, versus the observed NAM. These are otherwise very weak signals of an ensemble mean which might be very sensitive to a few ensemble members. I do not think this figure is enough to fully claim that a negative surface NAM would have occurred without the SSW.

We've added MERRA2 to Figure 3. It is indeed clear that the January 4th forecasts do not capture the positive anomaly in February. But it is also clear that the behavior among the forecasts for the first four weeks after the SSW supports the conclusion that the SSW itself exerted little impact on the surface circulation or on surface temperatures. That the positive geopotential height anomalies in the stratosphere fade without the observed tropospheric circulation indicates that the tropospheric circulation is a major driver of the downward propagation and duration of the event.

See the below plots of the lower-tropospheric geopotential height anomalies (with MERRA2 in red). The forecasts are well-distributed about the average, as may be expected from an average over 21 members, so we think the forecast average is indeed representative and not dominated by a few members.

The surface geopotential in the standard forecasts and forecasts with scrambled stratospheric initial conditions behave similarly, with generally positive anomalies throughout the first four weeks and numerous members capturing a 2+ standard deviation anomaly 2-6 weeks after initialization. The forecasts with scrambled tropospheric and atmospheric initial conditions behave similarly to one another. As this confirms the analysis of Figure 3 in the manuscript, we are omitting it for brevity.

Please also see the new discussion of Hitchcock and Simpson, and the method of relaxing the stratosphere to observations, in the new version of the manuscript. We do not think this method is easy to interpret, as it bakes-in any upward coupling effect into the prescribed stratospheric variability.

L237-241: This is where I am particularly concerned at the distinction between causality and predictability. The Texas cold wave was an extreme synoptic event taking place on a timescale of only a few days. I do not think this would be predictable at all at this lead-time, given it might require a perfect interaction of stratospheric forcing (or otherwise) with other subseasonal phenomena (the MJO was active during this time, for example). As it stands, to me this analysis just says that this model cannot predict an extreme temperature event 5 weeks ahead of time... which is not surprising. If the standard forecasts predicted the cold wave, and those with the scrambled stratospheric ICs also did, then I would agree that the stratospheric forcing was not important. But since neither predicted the cold wave, then how can we conclude the stratosphere was not important? A better approach might be to find the lead-time at which the standard forecast was able to predict the event, and then perform these experiments. As it stands, the conclusion I take from this is “subseasonal prediction of the Texas cold wave was not increased by the occurrence of the SSW”. However, I do agree with the authors that many were far too quick to attribute causality at the time.

Looking at Fig. 3a, there is no substantial signal of a negative NAM through the depth of the troposphere around the time of the Texas cold wave, as was observed – so it is not surprising to see an absence of the surface thermal signal.

We don't think there is necessarily such an obvious distinction between causality and predictability. As you say, the extreme cold “might require a perfect interaction of stratospheric forcing (or otherwise) with other subseasonal phenomena”, so the inability of a forecast to produce the extreme cold after an SSW may be a sign that the causal pathway to produce

extreme cold is tenuous and requires the interaction of many more processes than just the SSW. Alternatively, as you note, it raises the question of whether the model could ever produce such an event.

So we agree that this analysis was unsatisfying before, especially as there was little differentiation between the forecasts and we did not show whether the model could actually capture the event. We added a second set of forecasts initialized on February 8th, 2021 - see our first summarized major change. We believe it more rigorously supports the conclusion that this SSW did not likely drive the extreme cold over North America, especially as the standard forecast initialized on February 8th predicts (nearly) record cold.

Minor comments:

L14: are Arctic SSWs 'rare'? (same comment for L53) Once every other winter is relatively common, to my mind, in comparison with e.g., Antarctic SSWs. Further, the text here should specifically state Arctic SSWs.

From our perspective, it is rare as it is less common than a frontal system, a hurricane, or a severe tornado, and it often elicits excitement in the research community. "Rare" is a spectrum, but compared to other extreme events, we feel it is apt.

L15-17: I found this sentence hard to understand. Additionally, there seems to be some contradiction between the statements for "North America" and "Eastern United States", while I find the statement that there is 'stormy' weather in Eurasia perplexing... at least in Europe, it is NAO+/strong polar vortex conditions associated with 'stormy' weather.

Domeisen and Butler (2020) show that a weak vortex is associated with cold extremes over most of Europe, flooding in southern Europe, and extreme weather over the Atlantic.

We've revised this sentence to state "warm temperatures in Northeastern Canada and cold, stormy weather throughout Europe and the United States" to more accurately reflect the weak vortex state. Deng et al. (2013) show that the temperatures in North America are slightly different than the above composite - this is why we originally noted warmth over North America generally, but stormy weather in the Eastern United States. In general there are some

Editorial note: The above figure is from "Stratospheric drivers of extreme events at the Earth's surface" by Domeisen and Butler (2020), under a Creative Commons Attribution 4.0 license: <https://creativecommons.org/licenses/by/4.0/>

differences between weak vortex/negative NAM, we just did not take enough care in our characterization.

L31: Is it the database which is standardized, or the forecast? I suggest “standardized databases of subseasonal forecasts.”

Yes, this is less ambiguous than what we originally wrote.

L44: “could”... perhaps “can”?

Thanks, we agree this is more accurate.

L48: Suggest “associated with breaking planetary waves”

Yes, this is more clear - we’ve added “breaking”.

L49: Suggest changing “producing” to “known as”. It might be worth mentioning something about what constitutes a major event.

That’s more clear - we’ve changed this to “known as”. As “minor” SSW’s are not classified objectively, we’re not sure it’s worth discussing them - especially as they do not create a barrier to planetary wave activity in the same way as a “major” SSW.

L53: It is alluded to but maybe it could be more clearly stated that the wave activity need not be anomalous, e.g. Birner and Albers 2017 SOLA, de la Camara et al. 2018 J Clim

We’ve clarified this to say “otherwise normal wave energy”.

L62-63: I am not sure I understand “storminess increases the variability”.

We’ve removed this. Our thought was that as cold/warm frontal systems drive major changes in temperature over small geographic regions, their presence should tend to increase variability and therefore negatively impact forecast skill, but without a solid reference we don’t feel it’s worth making this statement.

L65: What is the “scientific novelty”? SSWs are not novel.

The Oxford dictionary defines “novel” as “New or unusual in an interesting way.” We think SSWs are unusual, as they are a rapid deceleration of an otherwise-stable vortex.

L79: “often”... is “usually” better?

Yes, thank you.

L84: I think this sentence makes sense without “the meteorology in”

Agreed, we’ve removed this.

L115: The absence of a notable impact within 2 weeks of the SSW occurrence seems within expectation, since it is broadly dominated by the timescale for the tropospheric NAM (e.g., Simpson et al. 2011 <https://doi.org/10.1029/2011GL049304>) and usually not long enough after the SSW for a typical downward impact to be noted (e.g., Hall et al. 2020 JGR <https://doi.org/10.1029/2020JD033881>). Some of the preconditioning of the vortex before the SSW itself may have played a role though (lower-mid stratospheric NAM was negative before 5 January) which is perhaps worth noting.

We’ve added the reference to Hall et al. (2020) - we understand the concept behind the tracking algorithm, but it is rather complicated. Maybe more importantly, it characterizes the descent of the maximum polar cap height anomaly, so it is somewhat conservative on the lag by choosing the time of peak impact. We think the experimental forecasts show that the coupling for this particular SSW was probably bidirectional, which would not have been concluded from the geopotential height anomalies in a typical simulation/reanalysis.

L170: Reference to ‘anomalies’ is needed, i.e., “polar-cap average geopotential height anomalies are similar to the NAM index, as both diagnose the anomalous geopotential slope”.

Thanks, we’ve added “anomalies” for clarity to all occurrences.

L185: It is very interesting that the troposphere may have enhanced the downward propagation. Analysis of the wave activity/eddy heat flux may show this was important – the eddy heat flux did remain high in the lower-stratosphere following the SSW onset. But probably beyond the scope of this study.

We think that a fixed eddy/fixed zonal mean modeling framework could more directly tease this out, but that is well beyond the scope of this paper.

L249-250: Yes, but there are many ways in which the atmosphere can produce cold conditions in Texas – not necessarily in a flow evolution which resembles what happened in 2021.

We decided to expand the averaging region to all North American land areas (except for the archipelago of the Northwest Territories), to illustrate the robustness of the extreme cold and its continental scale. We think the new forecasts provide convincing evidence that the SSW was not necessary for the simulation of the extreme cold.

L266: Please state how anomalies were standardized.

We have added this to the “Methods” section. They are standardized the conventional way - by subtracting the mean and dividing by the standard deviation.

L288: “Insensitive” to what extent? As it is a small domain, surely it becomes sensitive at some point?

See above - we expanded the averaging region to the North American landmass. The cold anomaly over the continent was extremely widespread, although the peak -16C anomalies were concentrated over Texas and Oklahoma.

L326: There was a minor SSW involving a significant vortex deceleration during late Jan 2017, but yes – there was no major event, and the vortex was strong on the date noted.

We’ve noted this in the text.

Figure 3: the very thin negative anomalies near the surface in the first few days of the Standard forecast are dubious to me.

Yes, these looked suspect! We identified an issue with the postprocessing of this forecast - we were not properly masking out NaNs during cosine weighting of the polar cap average (e.g., areas where the interpolated output was below the surface pressure; we were not removing these grid points’ cosine weights). We’ve fixed it, and it has fixed the surface anomalies such that they match those in the scrambled stratospheric initial conditions forecast. This does not affect the other forecasts, as they were processed using a newer version of code.

Review of “Limited surface impacts of the January 2021 sudden stratospheric warming” by Richter et al.

The study compares the contributions of the stratosphere versus the troposphere and surface boundary conditions to extratropical surface weather following the Stratospheric Sudden Warming (SSW) of January 2021. The causal role of each component is assessed using targeted experiments in a dynamical prediction model, where the initial conditions of a given component are set to observations while the others are "scrambled". The results argue for a limited influence of the SSW on the surface conditions that followed, particularly in Texas which experienced severely cold weather.

The approach of using a dynamical prediction model to assess the causal impact of a SSW on surface weather is technically sound. However, I found two issues with the study which may impact its publication in this journal. The first is that I'm unsure whether there is sufficiently strong evidence to support some of the claims, but this could potentially be addressed with some additional analysis. The second issue, following the stated aims and scope of this journal, is that I'm unsure this study represents a sufficiently 'important scientific advance of significant interest to specialists'. This could potentially be overcome if the authors frame their results a bit differently. These issues are explained in further detail below to help the authors improve their manuscript and help the editor reach a decision.

General comments:

1. I'm unsure whether there is sufficiently strong evidence to support some of the claims made in this study. It seems to me that in order to use the scrambled experiments to assess causality of the event, the standard/control forecast must first be able to capture the event itself. Otherwise, the scrambled experiments are isolating the causal influence of a different event. Figure 1b shows that the standard forecast captures an important fraction (50%) of the spatial pattern of temperature anomalies during the two weeks following the event. However, the fraction becomes progressively worse later in the forecast (Fig. 2b) to the point where the standard forecast does not capture the same event as observations (Fig. 4). This issue could potentially be addressed by repeating the same analyses except using forecasts initialised closer to the dates of the anomalies shown. This is because the goal of this study is not to demonstrate the ability of the model to forecast the observed surface weather at long-lead times but rather to establish the causal role of the stratosphere versus the other components.

We agree with your general sentiment - we've done two more forecasts initialized less than one week before the event to eliminate any ambiguity in the interpretation. The standard forecast predicts an event comparable to the second-coldest ever in the MERRA2 reanalysis, while a forecast with scrambled tropospheric initial conditions predicts an event comparable to the second-warmest ever in the MERRA2 reanalysis. We think this shows the model is capable of simulating extreme North America-wide cold and that the SSW likely did not drive the extreme cold in February.

An event highly sensitive to previous/initial conditions implies a tenuous causal relationship to one particular event, so we do not think there is such a clear distinction between predictability and causality.

2. As it stands, I'm unsure whether this study represents a sufficiently 'important scientific advance of significant interest to specialists'. There have been several other studies in specialist journals which explored why some SSWs couple with the troposphere while others do not, sometimes called 'propagating' versus 'non-propagating' SSWs [Gerber et al., 2009, Jucker, 2016, Karpechko et al., 2017, Nakagawa and Yamazaki, 2006, Rao et al., 2020, White et al., 2019]. To be fair, most of these studies assume that similar signed NAM anomalies in the troposphere and stratosphere following SSWs imply a causal influence from the stratosphere which is not necessarily the case as discussed here. However, Hitchcock and Simpson [2014] isolated the causal role of the stratosphere using stratospheric nudging experiments in a state-of-the-art climate model and showed that SSWs tend to shift the distribution of tropospheric weather favouring some conditions over others (see their Fig. 6). Thus, the results presented here are not really surprising because we expect that a large fraction of the stratospheric influence on the troposphere will be masked by internal tropospheric variability [Gerber et al., 2009]. Finally, the fact that this specific SSW event received a lot of media attention strikes me as insufficient motivation for specialists and more interesting to the general public.

I think the authors could frame their results differently, and thus increase the impact and novelty of their study, by arguing for the feasibility of their 'scrambling' approach for real-time attribution of events using operational forecasts models. To my knowledge, the scrambling approach has only been used to assess the causal role of the stratosphere in an idealized model [Gerber et al., 2009] and the results presented here could potentially be the first to demonstrate its use in a realistic setting. Since there is currently a lot of interest in 'windows of opportunity' during S2S forecasts, particularly from stratospheric sources, encouraging prediction centers to implement this type of analysis could be useful for understanding and improving forecast skill. The scrambling approach could be complementary to and easier to implement than the nudging approach which is currently being explored as part of the SNAPSI effort.

We somewhat disagree - we don't think Hitchcock and Simpson cleanly isolated a causal relationship, unless the stratosphere can be assumed to be fully isolated from tropospheric variability. Any upward coupling from the troposphere (Fig. 3) will be baked-in to the prescribed stratospheric variability in "perfect stratosphere" model experiments. This makes the interpretation of such experiments somewhat ambiguous.

The stratospheric state may have an impact on the troposphere, but here our focus is on the SSW itself as a distinct event from any stratosphere-troposphere coupling afterward - something that can be studied with the initial condition approach, but not the approach of nudging either the troposphere or stratosphere throughout the event. We've added a discussion of this to the "Discussion" section, in particular stating, "The ambiguity inherent to such a "perfect stratosphere" simulation makes it difficult to determine whether it is the SSW, or the

tropospheric circulation at SSW onset, or stratosphere-troposphere feedbacks, that lead to the negative NAM at the surface.”

Further, we think that public and media interest is excellent motivation. Public and media interest is generally an indication that an event had a material impact on lives and livelihoods. With the myriad ways something could be studied, public interest is a useful guide to align research toward the particular questions for which society desires answers.

As specialists have already made qualified attribution statements in scientific journals, without the benefit of experimentation, we think it is especially relevant for specialists. In particular, as this event has all the hallmarks of downward coupling based on an analysis of the polar cap geopotential heights, it raises the question of whether an examination of the polar cap geopotential height is a useful assessment of the dynamics without the aid of model experiments.

We've added references to White et al. (2019) and Jucker (2016) to the discussion and note that while this event was stereotypically downward propagating in its presentation, the event itself does not propagate downward without a particular tropospheric state at initialization. White et al. (2019) argue that a strong upward EP flux is a common tropospheric precursor to downward coupling. Their Figure 4 shows that most of this strong EP flux dissipates in the upper troposphere - which is necessarily associated with a negative tropospheric NAM as the drag (heat flux) associated with the dissipation weakens the equator-to-pole temperature gradient. It is unclear whether the negative tropospheric NAM at positive lags is actually a byproduct of the heat flux (it would happen without the SSW), or whether it creates the conditions for downward propagation. Jucker makes the case that the duration of the lower stratospheric NAM may also be a critical factor.

We think Fig. 3b-d here show that the tropospheric NAM would develop much the same over the next month, and that it induces the lower stratospheric NAM; but that feedbacks between the SSW and troposphere sustained the stratospheric and tropospheric NAM signal beyond what would have otherwise occurred. To us, “stratosphere-troposphere feedbacks”, or resonance, would seem a more useful model of this SSW's impacts on the surface than “downward propagating” or “non-downward-propagating”.

Specific comments:

1. L50-53: A good reference here would also be Dunn-Sigouin and Shaw [2020].

Thanks, we've added this reference.

2. L56-63: I think the authors should also refer to the studies cited above about propagating versus non-propagating SSWs.

We've cited White et al. and Jucker, but instead in the discussion where we note this particular event would satisfy the “downward propagating” categorization.

3. L131-141, Fig. S2 and Table S1: considering major comment 1 above, I don't think showing how poorly other models forecast the anomalies at long lead-times helps to support the results.

By the same coin, we think showing that this model outperformed others at forecasting the anomalies at these lead times builds confidence in the model and its usefulness for studying this event.

4. L183-185: a relevant citation here would be Plumb and Semeniuk [2003]

Thanks, we agree this is a great reference.

5. L195-197: I think this could be rephrased for clarity. The authors could also mention explicitly that their results are opposite those published previously about this event.

We have revised this, and now note it is opposite to two papers published previously: "We note this is contrary to analyses of observed geopotential heights, which have argued that this SSW likely had a strong projection onto surface weather".

6. L207 and elsewhere: I suggest replacing 'variance' with 'spatial variance' for clarity.

Agree, this is more clear - we've added "spatial" to all of its occurrences.

7. Figs 1,2 and 3: I think it would be helpful to include a significance measure of the anomalies shown. The forecast could have a similar mean to observations but could also be highly uncertain and thus have low significance.

We've added statistical significance to Figures 1-3, and have described how this was performed in the "Methods" section.

8. Fig. 3: Could the authors include a panel with observations for comparison as in the other figures?

Yes, this is a good idea - we've added MERRA2 as a verification, and have also used it for Figures 1-2 so that there is verification over the oceans.

9. Analysis: Have the authors considered looking at other weather relevant variables like precipitation? Would the results be similar?

Precipitation skill drops off quickly after the first week, so we chose to focus on temperature and geopotential height, which tend to have more sustained skill. We did briefly consider sea-level pressure, but see the response to the first reviewer, above - in the same way as surface temperature, the forecasts with scrambled stratospheric conditions are most similar to the standard forecasts, while the forecasts with scrambled tropospheric conditions are least similar.

10. Analysis: How did the authors calculate the squared spatial correlation coefficient? Is this similar to the anomaly correlation coefficient (ACC, e.g., equation 2 in Domeisen et al. [2020]). Have the authors looked at other commonly used skill measures of their forecasts? Would the results be consistent?

We've added a section to "Methods" detailing the calculation - it is just the ACC, squared and multiplied by the sign of the ACC, to directly communicate the fraction of variance shared between the forecast and the verification.

The RMSE, in terms of the relationship among the forecasts, does not provide any different interpretation - see the table below.

Table: Root-mean-square difference between each forecast and MERRA2, and root-mean-square difference between each forecast and the standard forecast, for weeks 1-2 (weeks 3-4), in units of degrees Celsius.

Forecast	RMS difference (MERRA2)	RMS difference (standard forecast)
Standard forecast	2.0 (2.8)	N/A
Scrambled tropospheric i.c.'s	4.8 (3.1)	4.7 (1.8)
Scrambled stratospheric i.c.'s	2.2 (3.0)	0.8 (1.6)
Scrambled atmospheric i.c.'s	4.7 (3.9)	2.6 (4.9)

We found the squared ACC useful for scoring how much variance is explained by the different initial conditions. We would say the ACC itself is not central to the results of this study, though, as the forecasts exhibit markedly different behavior.

Review of NCOMMS-21-14662A “Limited surface impacts of the January 2021 sudden stratospheric warming” revision 1

This is my second review of this manuscript. I am grateful to the authors for their thorough consideration of, and clear response to, both mine and the other reviewers’ comments on the first version. As such, I believe the manuscript is much improved and should be suitable given a further revision as outlined below. Please note that no line numbers were supplied so I have lifted specific quotes out.

Major comments:

General comment on cold weather in North America following SSWs: It’s important to note that several studies have suggested that the coldest weather in North America is not associated with major SSWs [Lee et al. 2019 GRL <https://doi.org/10.1029/2019GL085592>, Kretschmer et al. 2018 npj <https://doi.org/10.1038/s41612-018-0054-4>]. Therefore, the findings of this study do support previous work (rather than just countering any early qualitative analysis made after the 2021 SSW).

Figure 1: how much of this apparent effect comes from week 1, though? Scrambling the tropospheric ICs will by design catastrophically impact the first week surface conditions due to e.g. timescale of NAM/NAO (~1 week; and this is just about visible in Fig. 3c where the NAM starts out positive) before other forcing (e.g. from the stratosphere) can really impact the predictions. I like the analysis in 2-week chunks considered in Fig. 2, but find it a little hard to conclude anything from this because of week 1. I suggest that the analysis here is truncated to solely week 2; nothing should be lost from the conclusions.

Figure 4: I appreciate the revision carried out following my comments on the earlier version. However, in a similar vein to my comment on Figure 1, I am not sure this answers the question. February 8th is too close to the start of the verification period (synoptic timescales). I don’t think scrambling the tropospheric ICs and getting unusually warm conditions is evidence that the stratosphere would have otherwise caused this; on these timescales it’s more influenced by the initial state – a greater lag would be needed to see if there was an effect from the stratosphere. The focus on a very short-range forecast here also runs counter to the main focus of the paper on subseasonal forecasting. It would also be informative to show a map, like in Figs 1 & 2, of the temperature anomalies during this period – supporting information will suffice.

Figure S1: The forecasts with the scrambled tropospheric ICs show a recovery of the vortex soon after the warming – with the verification running much closer to the weak end of the ensemble. Hence the stratosphere was evidently not in an isolated, constrained state post-SSW, and this brings out the idea of two-way interaction between the troposphere and stratosphere. I think this evidence could be emphasized in the paper to support the conclusions.

Minor comments:

“SSWs are predictable up to two weeks in advance” – perhaps also mention skill on the seasonal scale exists (e.g. Scaife et al. 2016 <https://doi.org/10.1002/asl.598>)

“Operational forecast models often use data assimilation” – should ‘often’ be ‘generally’?

SSWs as “rare” events – I will accept the authors’ decision on this, but I still believe the comparison with e.g. the frequency of a frontal system (as in the reviewer response) is unfair. The large spatial scale of an SSW invokes a longer timescale, so I would argue that they are thus relatively common in the Arctic (especially recently, it is hard to argue 2018, 2019 and 2021 is ‘rare’). This is just food for thought.

Reviewer #2 (Remarks to the Author):

Thank you for clearly and effectively responding to my comments. I am now convinced that the claims are supported by the evidence shown. I have three additional minor comments which should be addressed before publication.

1) After rereading the methods section of manuscript, I noticed that the scrambling/nudging procedure in the experiments could effect the results.

First, the tropopause changes as a function of height and latitude (e.g, ~300hPa in the Arctic and ~150hPa in the tropics) but the nudging/scrambling procedure is only a function of height. Thus, a portion of each layer relevant to the dynamics of the coupling could be missing in each experiment.

Second, the heights at which the nudging/scrambling are performed are different for the stratosphere and troposphere only experiments and these choices could be an issue. If I understand correctly, the troposphere scrambling experiment nudges down to 15km, which omits a component of the lower-stratosphere in the initial condition. Likewise, the stratosphere scrambling experiment nudges up to 12km which includes a portion of the stratosphere in the initial condition.

I think the height levels of the nudging/scrambling procedure and the uncertainty related to these choices should be mentioned clearly in the main text. Even better would be to perform additional sensitivity experiments but I realise this could be difficult and so I don't think it necessary.

2) I think the usage of the word 'resonance' in the new discussion section is a bit confusing and that simply referring to feedbacks is sufficient. In the SSW literature, resonance refers to multiple reflections in a cavity composed of turning points where the index of refraction equals 0 according to linear wave theory (see for example 'Planetary waves in the Extratropical Stratosphere' by Alan Plumb). Thus, it has a specific meaning which is not what the authors are referring to (I think).

3) Could the author's change the significance hatching in the figures to a light grey? White is hard to see.

To the reviewers,

Thank you for your additional comments. We've made several major changes to the manuscript to address your concerns, including the contention that our results support the role of vortex stretching/wave reflection in driving extreme cold in the United States (rather than SSW's). We weren't previously familiar with this area of research, but it is clear that it is important to address it as much as possible. Below is a summary of the major changes to the manuscript:

1. **Addition of February 1st and 8th forecasts with scrambled stratospheric initial conditions.** As noted by reviewer #2, Kretschmer et al. 2018 (and recently, Cohen et al. 2021) show wave reflection during polar vortex stretching events is associated with anomalous cold over North America. This process is distinct from the dynamics of SSW's, hence Reviewer #2's statement that "the findings of this study do support previous work". We wanted to ensure that this was the case, so we performed these additional forecasts as a quick check of whether such dynamics preceded the February event, and whether denying the mechanism from occurring would change the outcome of the event.
2. **Expanded analysis and discussion of the wave dynamics of the February extreme cold.** Accordingly, we've added a new figure (Fig. 5) that displays the 100 hPa geopotential height anomalies and vertical wave activity flux preceding the event, as well as zonal (Fig. 6) and meridional (Fig. 7) projections of the wave activity flux. We show that the standard forecast captures much of the observed behavior in the wave dynamics, including the SPV stretching/wave reflection identified by Kretschmer et al. and Cohen et al and referred to by Reviewer #2. Further, we show that the forecasts with scrambled stratospheric initial conditions produce weak SPV stretching and no wave reflection, yet still predicts the same extreme cold as the standard forecast (Fig. 4). We believe this is evidence that the SPV stretching/wave reflection process is irrelevant for this particular cold air outbreak. This is plausible based on an analysis of MERRA2 alone. Using streamlines based on physically-sound wave activity flux vector scaling, we show that the reflected wave activity at 100 hPa in MERRA2 never reaches the troposphere.

We realize this is not a trivial amount of additional material. However, we were not comfortable suggesting that our work supports previous findings on the role of vortex stretching and wave reflection without some examination. We hope the reviewers and editor recognize why we are taking great care and time in returning the manuscript, given the stakes and the issues we raise. We hope we have addressed your primary concerns about predictability/lead time of the February forecasts, and that you find our treatment of the stratospheric dynamics and extreme cold event coupling well-reasoned.

Sincerely,
The Authors

Response to Reviewer #1

1) After rereading the methods section of manuscript, I noticed that the scrambling/nudging procedure in the experiments could effect the results.

First, the tropopause changes as a function of height and latitude (e.g, ~300hPa in the Arctic and ~150hPa in the tropics) but the nudging/scrambling procedure is only a function of height. Thus, a portion of each layer relevant to the dynamics of the coupling could be missing in each experiment.

Second, the heights at which the nudging/scrambling are performed are different for the stratosphere and troposphere only experiments and these choices could be an issue. If I understand correctly, the troposphere scrambling experiment nudges down to 15km, which omits a component of the lower-stratosphere in the initial condition. Likewise, the stratosphere scrambling experiment nudges up to 12km which includes a portion of the stratosphere in the initial condition.

I think the height levels of the nudging/scrambling procedure and the uncertainty related to these choices should be mentioned clearly in the main text. Even better would be to perform additional sensitivity experiments but I realise this could be difficult and so I don't think it necessary.

Regarding the latitude dependency, we think the most important coupling dynamics occur in the extratropics, which is why the height at which the nudging tapers in the scrambled stratospheric initial condition spin-up simulation coincides with the approximate height of the extratropical tropopause at 9 km.

We made the choice to taper the scrambled stratospheric initial condition spin-up simulation starting at 9 km so that the entire troposphere would be correctly initialized. The nudging timescale tapers exponentially such that by 10 km it is already extremely long. There is no way to abruptly taper the nudging without incurring instabilities, so we had to make some sort of tradeoff. However, the important component of any downward coupling by the SSW is the SSW itself, which is not initialized in the scrambled stratosphere forecasts and is initialized in the scrambled troposphere forecasts (Fig. S1).

In terms of the tapering heights, we did perform a sensitivity analysis of the scrambled stratospheric initial condition forecasts - see below (on the next page). When we taper the nudging from 15 km to 20 km, instead of 9 km and 12 km, we see identical behavior in weeks 1-2 and very similar behavior in weeks 3-4. However, the forecasts with a 15 to 20km taper are slightly more similar to the standard forecast with weaker cold over Siberia. We believe this means that while the lower stratospheric i.c.'s may mediate tropospheric weather after this SSW, they definitely do not "drive" it. This is not a totally satisfactory answer to your concern, but we hope it illustrates that including more or less of the lower stratosphere will not produce a notably different interpretation of the event.

As far as the tapering not overlapping, our concern was that the lower stratospheric circulation was not associated with the SSW itself, but was instead forced by the troposphere. As our intent was to assess the direct impact of the SSW in isolation, we believed this to be the cleanest separation. We essentially left the lower stratosphere orphaned because of its role as a coupling layer, and our thought that properly initializing it in either experiment might tilt the results too much in favor of either the troposphere or the SSW. We now note our intent with the nudging tapering heights in the methods section so that readers do not have to guess why we chose different heights.

2) I think the usage of the word 'resonance' in the new discussion section is a bit confusing and that simply referring to feedbacks is sufficient. In the SSW literature, resonance refers to multiple reflections in a cavity composed of turning points where the index of refraction equals 0 according to linear wave theory (see for example 'Planetary waves in the Extratropical Stratosphere' by Alan Plumb). Thus, it has a specific meaning which is not what the authors are referring to (I think).

Yes, we do not mean to cause any confusion and have changed this to “feedback process”. Resonance is a particular type of feedback process, and without a particular theoretical argument we agree that we should leave it as general as possible.

3) Could the author's change the significance hatching in the figures to a light grey? White is hard to see.

Sure, we've updated the figures to have darker hatching so that it is easier to identify the non-significant regions.

Response to Reviewer #2 comments

This is my second review of this manuscript. I am grateful to the authors for their thorough consideration of, and clear response to, both mine and the other reviewers' comments on the first version. As such, I believe the manuscript is much improved and should be suitable given a further revision as outlined below. Please note that no line numbers were supplied so I have lifted specific quotes out.

Major comments:

General comment on cold weather in North America following SSWs: It's important to note that several studies have suggested that the coldest weather in North America is not associated with major SSWs [Lee et al. 2019 GRL <https://doi.org/10.1029/2019GL085592>, Kretschmer et al. 2018 npj <https://doi.org/10.1038/s41612-018-0054-4>]. Therefore, the findings of this study do support previous work (rather than just countering any early qualitative analysis made after the 2021 SSW).

Lee et al. shows that the probability of occurrence of cold air outbreaks is highest in the Alaskan Ridge regime, though it is also elevated in the Arctic High regime. The former is not associated with variations in vortex strength, while the latter is a strong function of vortex strength. This study does not support the claim that the *coldest* weather in North America is not associated with major SSW's - it instead supports the claim that generally cold weather is only *sometimes* associated with a weak vortex.

Likewise, the cluster analysis in Kretschmer et al. shows that only about 10% of the time, the regime associated with cold air outbreaks in North America is associated with an SSW. Therefore, while there are many cold events *not* associated with an SSW, we think it would be incorrect to state the "coldest weather in North America is not associated with major SSWs".

The 100hPa circulation regime on February 4th, one week before the extreme cold, resembles the 100 hPa Alaskan High pattern in Lee et al. and the Cluster 4 pattern in Kretschmer et al. associated with the greatest risk for extreme cold. See below, cf. Fig. 9 of Wright et al., Fig. 4 of Lee et al, and Fig. 1 of Kretschmer, and see also the new Fig. 5 in this manuscript. Note also the probability of the Alaskan high event is elevated about one month after a weak vortex (Fig. 2 of Lee et al.).

We've added a discussion of the evolution of stratospheric geopotential height (Fig. 5) and wave dynamics before and during this event (Fig 5-7). When we shut down the wave reflection/vortex stretching process in the 12 days before the event by scrambling the stratospheric initial conditions, the surface temperature forecast of near-record cold over North America is unchanged (Fig. 4). This is probably because, as we note in the text, the reflected waves in the stratosphere did not reach the troposphere during the event, anyway (Fig. 6-7).

Figure 1: how much of this apparent effect comes from week 1, though? Scrambling the tropospheric ICs will by design catastrophically impact the first week surface conditions due to e.g. timescale of NAM/NAO (~1 week; and this is just about visible in Fig. 3c where the NAM starts out positive) before other forcing (e.g. from the stratosphere) can really impact the predictions. I like the analysis in 2-week chunks considered in Fig. 2, but find it a little hard to conclude anything from this because of week 1. I suggest that the analysis here is truncated to solely week 2; nothing should be lost from the conclusions.

Below is the week 2-only composite - we don't think it is substantially different (quantitatively or qualitatively) from the week 1-2 composite.

We agree the adjustment time of tropospheric variability is approximately one week. However, the forecasts are scrambled by spinning up the stratosphere and letting the troposphere adjust (the Hitchcock and Simpson approach to examining the SSW impacts), rather than randomizing the troposphere at initialization. The troposphere at initialization has already “felt” the stratosphere and is in balance with it. We’ve added a discussion of this to our description of the model setup in the main text. But more importantly, as you state, scrambling the troposphere leads to a very different pattern of weather - which is a key point of this study. The stratosphere in isolation cannot be said to govern surface weather, while the troposphere in isolation seems to capture the most important variability, at least during this event.

Figure 4: I appreciate the revision carried out following my comments on the earlier version. However, in a similar vein to my comment on Figure 1, I am not sure this answers the question. February 8th is too close to the start of the verification period (synoptic timescales). I don’t think scrambling the tropospheric ICs and getting unusually warm conditions is evidence that the stratosphere would have otherwise caused this; on these timescales it’s more influenced by the initial state – a greater lag would be needed to see if there was an effect from the stratosphere. The focus on a very short-range forecast here also runs counter to the main focus of the paper on subseasonal forecasting. It would also be informative to show a map, like in Figs 1 & 2, of the temperature anomalies during this period – supporting information will suffice.

As the troposphere in the spin-up run has “felt” the correct stratospheric variability, we think it is more nuanced than the troposphere being randomized at initialization. Our initialization procedure is very similar to the Hitchcock and Simpson methodology for examining the impacts of SSWs. Any deterministic effect from the stratosphere is by construction included in the tropospheric initial condition in the scrambled tropospheric initial conditions - it is the effects due to a *particular* tropospheric state that are excluded from the initial conditions. This does indeed suggest that a particular tropospheric state might be necessary for the stratosphere to impact the surface weather in the expected ways described in the statistical analyses in Lee et al. [2019] and Kretschmer et al. [2018]. We believe this expands the view provided by existing studies, and leaves open, as we note in the discussion of the vertical structure of geopotential height, the need for bidirectional coupling to explain longer lead-time variability. However, for the initial month, the troposphere, in total isolation, seems to be sufficient to explain the general pattern of surface weather.

We’ve added a figure with surface temperature maps to the supporting information. In general, the standard forecast and forecast with scrambled stratospheric initial conditions reproduce the cold weather down to 40N, and while they do not produce the extreme magnitude of cold weather over Texas, they do predict anomalous cold all the way to the US/Mexico border. Additionally, the observed warmth over Greenland is instead shifted to the southwest. Both of these errors prevent the forecasts from producing continent-wide surface temperatures as cold as observed (Fig. 4). The forecast with scrambled tropospheric initial conditions instead shows broad warmth throughout the United States and Canada.

Figure S1: The forecasts with the scrambled tropospheric ICs show a recovery of the vortex soon after the warming – with the verification running much closer to the weak end of the ensemble. Hence the stratosphere was evidently not in an isolated, constrained state post-SSW, and this brings out the idea of two-way interaction between the troposphere and stratosphere. I think this evidence could be emphasized in the paper to support the conclusions.

That is a great point, we've added this to the discussion of the geopotential height anomalies, as this indeed illustrates the importance of two-way coupling in sustaining the circulation regime.

Minor comments:

“SSWs are predictable up to two weeks in advance” – perhaps also mention skill on the seasonal scale exists (e.g. Scaife et al. 2016 <https://doi.org/10.1002/asl.598>)

We are beyond the (suggested) reference limit of 70. Because Scaife et al. (2016) is a more extreme case that illustrates the connection between SSWs and the seasonal variability of the tropospheric circulation, we've decided to substitute Scaife et al. in place of Karpechko (2018).

“Operational forecast models often use data assimilation” – should ‘often’ be ‘generally’?

Yes, that's a fair revision. We can't really say “all”, but any advanced forecasting center uses DA.

SSWs as “rare” events – I will accept the authors' decision on this, but I still believe the comparison with e.g. the frequency of a frontal system (as in the reviewer response) is unfair. The large spatial scale of an SSW invokes a longer timescale, so I would argue that they are thus relatively common in the Arctic (especially recently, it is hard to argue 2018, 2019 and 2021 is ‘rare’). This is just food for thought.

Okay, you've sufficiently convinced us that there is a better way to frame them! We no longer refer to them as “rare” and instead describe them as “abrupt”. Perhaps the most noteworthy aspect of their prediction, especially on seasonal timescales, is that they are a dynamically-forced event that occurs over a short time period.

REVIEWERS' COMMENTS

Reviewer #1 (Remarks to the Author):

I thank the authors for their careful consideration of both mine and the other reviewer's comments, their clear responses, and additional work on the manuscript to include stretching and reflection analysis.

I now believe this is a much-improved manuscript with several novel results which will prompt much discussion and further research in the community. Thus, in my view, it deserves publication.

Minor comment:

L269 suggest changing "hottest" to "warmest".

Reviewer #2 (Remarks to the Author):

Thanks for answering my additional questions. It's also interesting to see the new results regarding wave reflection. I have a single optional comment below about the interpretation on lines 348-353 of the revised manuscript. However, I do not need to review this manuscript again.

An alternative interpretation for why downward pointing stratospheric E-P flux vectors occurred but did not effect the troposphere is that downward pointing EP-flux vectors do not necessarily imply downward wave propagation/reflection if the conditions are not linear (Plumb 2010, <https://doi.org/10.1002/9781118666630.ch2>). In other words, linear theory can be misleading.

Explicit evidence of processes leading to downward pointing stratospheric EP-fluxes, separate from wave reflection, were shown in idealized experiments by Dunn-Sigouin and Shaw 2018 (<https://doi.org/10.1175/JAS-D-17-0263.1>).

This supports your results so feel free to include this point in the paper if you want.

To the reviewers,

Thank you for your attention to this manuscript, we feel it is much improved from our initial submission.

Please find our responses to your minor comments, and the tracked-changes manuscript.

Thank you,
The authors

Response to Reviewer #1

I thank the authors for their careful consideration of both mine and the other reviewer's comments, their clear responses, and additional work on the manuscript to include stretching and reflection analysis.

I now believe this is a much-improved manuscript with several novel results which will prompt much discussion and further research in the community. Thus, in my view, it deserves publication.

Thank you again for all of your comments and insights, we appreciate them all.

Minor comment:

L269 suggest changing "hottest" to "warmest".

Agreed, and changed.

Response to Reviewer #2

Thanks for answering my additional questions. It's also interesting to see the new results regarding wave reflection. I have a single optional comment below about the interpretation on lines 348-353 of the revised manuscript. However, I do not need to review this manuscript again.

Thanks, we appreciate the time you took to help us improve this manuscript.

An alternative interpretation for why downward pointing stratospheric E-P flux vectors occurred but did not effect the troposphere is that downward pointing EP-flux vectors do not necessarily imply downward wave propagation/reflection if the conditions are not linear (Plumb 2010, <https://doi.org/10.1002/9781118666630.ch2>). In other words, linear theory can be misleading.

Explicit evidence of processes leading to downward pointing stratospheric EP-fluxes, separate from wave reflection, were shown in idealized experiments by Dunn-Sigouin and Shaw 2018 (<https://doi.org/10.1175/JAS-D-17-0263.1>).

This supports your results so feel free to include this point in the paper if you want.

Thanks, we will include the chapter, Planetary waves and the extratropical winter stratosphere, in our references, and now note here that “The wave activity flux may also be misleading if conditions were not linear”. We would rather keep it simple and just note that the assumption of linearity during an extreme event may not necessarily lead to a reliable interpretation.